# STOCHASTIC AUC MAXIMIZATION WITH DEEP NEURAL NETWORKS

**Mingrui Liu**
Department of Computer Science
The University of Iowa
Iowa City, IA, 52242, USA
`mingrui-liu@uiowa.edu`

**Zhuoning Yuan**
Department of Computer Science
The University of Iowa
Iowa City, IA, 52242, USA
`zhuoning-yuan@uiowa.edu`

**Yiming Ying**
Department of Mathematics and Statistics
SUNY at Albany
Albany, NY, 12222, USA
`yying@albany.edu`

**Tianbao Yang**
Department of Computer Science
The University of Iowa
Iowa City, IA, 52242, USA
`tianbao-yang@uiowa.edu`

## ABSTRACT

Stochastic AUC maximization has garnered an increasing interest due to better fit to imbalanced data classification. However, existing works are limited to stochastic AUC maximization with a linear predictive model, which restricts its predictive power when dealing with extremely complex data. In this paper, we consider stochastic AUC maximization problem with a deep neural network as the predictive model. Building on the saddle point reformulation of a surrogated loss of AUC, the problem can be cast into a *non-convex concave* min-max problem. The main contribution made in this paper is to make stochastic AUC maximization more practical for deep neural networks and big data with theoretical insights as well. In particular, we propose to explore Polyak-Łojasiewicz (PL) condition that has been proved and observed in deep learning, which enables us to develop new stochastic algorithms with even faster convergence rate and more practical step size scheme. An AdaGrad-style algorithm is also analyzed under the PL condition with adaptive convergence rate. Our experimental results demonstrate the effectiveness of the proposed algorithms.

## 1 INTRODUCTION

Deep learning has been witnessed with tremendous success for various tasks, including computer vision (Krizhevsky et al., 2012; Simonyan & Zisserman, 2014; He et al., 2016; Ren et al., 2015), speech recognition (Hinton et al., 2012; Mohamed et al., 2012; Graves, 2013), natural language processing (Bahdanau et al., 2014; Sutskever et al., 2014; Devlin et al., 2018), etc. From an optimization perspective, all of them are solving an empirical risk minimization problem in which the objective function is a surrogate loss of the prediction error made by a deep neural network in comparison with the ground-truth label. For example, for image classification task, the objective function is often chosen as the cross entropy between the probability distribution calculated by forward propagation of a convolutional neural network and the vector encoding true label information (Krizhevsky et al., 2012; Simonyan & Zisserman, 2014; He et al., 2016), where the cross entropy is a surrogate loss of the misclassification rate. However, when the data is imbalanced, this formulation is not reasonable since the data coming from minor class have little effect in this case and the model is almost determined by the data from the majority class.

To address this issue, AUC maximization has been proposed as a new learning paradigm (Zhao et al., 2011). Statistically, AUC (short for Area Under the ROC curve) is defined as the probability that the prediction score of a positive example is higher than that of a negative example (Hanley & McNeil, 1982; 1983) . Compared with misclassification rate and its corresponding surrogate loss, AUC is more suitable for imbalanced data setting (Elkan, 2001). Several online or stochastic algorithms for

AUC maximization have been developed based on a convex surrogate loss (Zhao et al., 2011; Gao et al., 2013; Ying et al., 2016; Liu et al., 2018; Natole et al., 2018). However, all of these works only consider learning a linear predictive model. This naturally motivates the following question:

**How to design stochastic algorithms with provable guarantees to solve the AUC maximization problem with a deep neural network as the predictive model?**

In this paper, we make some efforts to answer this question. We design two algorithms with state-of-the-art complexities for this problem. Based on a surrogate loss of AUC and inspired by the min-max reformulation in (Ying et al., 2016), we cast the problem into a non-convex concave min-max stochastic optimization problem, where it is nonconvex in the primal variable and concave in the dual variable. This allows us to leverage the inexact proximal point algorithmic framework proposed in (Rafique et al., 2018) to solve stochastic AUC maximization with a deep neural network. However, their algorithms are limited for stochastic AUC maximization with a deep neural network due to three reasons. First, their algorithms are general and do not utilize the underlying favorable property of the the objective function induced by an overparameterized deep neural network, which prevents them from designing better algorithms with faster convergence. Second, these algorithms use a polynomially decaying step size scheme instead of the widely used geometrically decaying step size scheme in deep neural network training. Third, the algorithm in (Rafique et al., 2018) with the best attainable complexity only applies to the finite-sum setting, which needs to go through all data at the end of each stage and are not applicable to the pure stochastic setting.

To address these limitations, we propose to leverage the Polyak-Łojasiewicz (PL) condition of the objective function for AUC maximization with a deep neural network. The PL condition (or its equivalent condition) has been proved for a class of linear and non-linear neural networks (Hardt & Ma, 2016; Charles & Papailiopoulos, 2017; Zhou & Liang, 2017). It is the key to recent developments that prove that (stochastic) gradient descent can find a global minimum for an overparameterized deep neural network (Allen-Zhu et al., 2018; Du et al., 2018b). It is also observed in practice for learning deep neural networks (Li & Yuan, 2017; Kleinberg et al., 2018). From an optimization perspective, the PL condition has been considered extensively for designing faster optimization algorithms in the literature (Karimi et al., 2016; Reddi et al., 2016; Lei et al., 2017). However, there still remains **a big gap** between existing algorithms that focus on solving a minimization problem and the considered min-max problem of AUC maximization. It is a non-trivial task to leverage the PL condition of a non-convex minimization objective for developing faster primal-dual stochastic algorithms to solve its equivalent non-convex concave min-max problem. The main theoretical contributions in this paper are to solve this issue. Our contributions are:

- We propose a stochastic algorithm named Proximal Primal-Dual Stochastic Gradient (PPD-SG) for solving a min-max formulation of AUC maximization under the PL condition of the surrogated AUC objective with a deep neural network. We establish a convergence rate in the order of $O(1/\epsilon)$, which is faster than that achieved by simply applying the result in (Rafique et al., 2018) to the considered problem under the PL condition, i.e., $O(1/\epsilon^3)$ and $O(n/\epsilon)$ with $n$ being the size of training set.

- In addition, we propose an AdaGrad-style primal-dual algorithm named Proximal Primal-Dual Adagrad (PPD-Adagrad), and show that it enjoys better adaptive complexity when the growth of cumulative stochastic gradient is slow. This is the first time an adaptive convergence of a stochastic AdaGrad-style algorithm is established for solving non-convex concave min-max problems.

- We evaluate the proposed algorithms on several large-scale benchmark datasets. The experimental results show that our algorithms have superior performance than other baselines.

To the best of our knowledge, this is the first work incorporating PL condition into stochastic AUC maximization with a deep neural network as the predictive model, and more generally into solving a non-convex concave min-max problem. Our results achieve the state-of-the-art iteration complexity for non-convex concave min-max problems.

## 2 RELATED WORK

**Stochastic AUC Maximization.** Stochastic AUC maximization in the classical online setting is challenging due to its pairwise nature. There are several studies trying to update the model each

time based on a new sampled/received training data. Instead of storing all examples in the memory, Zhao et al. (2011) employ reservoir sampling technique to maintain representative samples in a buffer, based on which their algorithms update the model. To get optimal regret bound, their buffer size needs to be $O(\sqrt{n})$, where $n$ is the number of received training examples. Gao et al. (2013) design a new algorithm which is not buffer-based. Instead, their algorithm needs to maintain the first-order and second-order statistics of the received data to compute the stochastic gradient, which is prohibitive for high dimensional data. Based on a novel saddle-point reformulation of a surrogate loss of AUC proposed by (Ying et al., 2016), there are several studies (Ying et al., 2016; Liu et al., 2018; Natole et al., 2018) trying to design stochastic primal-dual algorithms. Ying et al. (2016) employ the classical primal-dual stochastic gradient (Nemirovski et al., 2009) and obtain $\widetilde{O}(1/\sqrt{t})$ convergence rate. Natole et al. (2018) add a strongly convex regularizer, invoke composite mirror descent (Duchi et al., 2010) and achieve $\widetilde{O}(1/t)$ convergence rate. Liu et al. (2018) leverage the structure of the formulation, design a multi-stage algorithm and achieve $\widetilde{O}(1/t)$ convergence rate without strong convexity assumptions. However, all of them only consider learning a linear model, which results in a convex objective function.

**Non-Convex Min-max Optimization.** Stochastic optimization of non-convex min-max problems have received increasing interests recently (Rafique et al., 2018; Lin et al., 2018; Sanjabi et al., 2018; Lu et al., 2019; Jin et al., 2019). When the objective function is weakly convex in the primal variable and is concave in the dual variable, Rafique et al. (2018) design a proximal guided algorithm in spirit of the inexact proximal point method (Rockafellar, 1976), which solves a sequence of convex-concave subproblems constructed by adding a quadratic proximal term in the primal variable with a periodically updated reference point. Due to the potential non-smoothness of objective function, they show the convergence to a nearly-stationary point for the equivalent minimization problem. In the same vein as (Rafique et al., 2018), Lu et al. (2019) design an algorithm by adopting the block alternating minimization/maximization strategy and show the convergence in terms of the proximal gradient. When the objective is weakly convex and weakly concave, Lin et al. (2018) propose a proximal algorithm which solves a strongly monotone variational inequality in each epoch and establish its convergence to stationary point. Sanjabi et al. (2018) consider non-convex non-concave min-max games where the inner maximization problem satisfies a PL condition, based on which they design a multi-step deterministic gradient descent ascent with convergence to a stationary point. It is notable that our work is different in that (i) we explore the PL condition for the outer minimization problem instead of the inner maximization problem; (ii) we focus on designing stochastic algorithms instead of deterministic algorithms.

**Leveraging PL Condition for Minimization.** PL condition is first introduced by Polyak (Polyak, 1963), which shows that gradient descent is able to enjoy linear convergence to a global minimum under this condition. Karimi et al. (2016) show that stochastic gradient descent, randomized coordinate descent, greedy coordinate descent are able to converge to a global minimum with faster rates under the PL condition. If the objective function has a finite-sum structure and satisfies PL condition, there are several non-convex SVRG-style algorithms (Reddi et al., 2016; Lei et al., 2017; Nguyen et al., 2017; Zhou et al., 2018; Li & Li, 2018; Wang et al., 2018), which are guaranteed to converge to a global minimum with a linear convergence rate. However, the stochastic algorithms in these works are developed for a minimization problem, and hence is not applicable to the min-max formulation for stochastic AUC maximization. To the best of our knowledge, Liu et al. (2018) is the only work that leverages an equivalent condition to the PL condition (namely quadratic growth condition) to develop a stochastic primal-dual algorithm for AUC maximization with a fast rate. However, as mentioned before their algorithm and analysis rely on the convexity of the objective function, which does not hold for AUC maximization with a deep neural network.

Finally, we notice that PL condition is the key to many recent works in deep learning for showing there is no spurious local minima or for showing global convergence of gradient descent and stochastic gradient descent methods (Hardt & Ma, 2016; Li & Yuan, 2017; Arora et al., 2018; Allen-Zhu et al., 2018; Du et al., 2018b;a; Li & Liang, 2018; Allen-Zhu et al., 2018; Zou et al., 2018; Zou & Gu, 2019). Using the square loss, it has also been proved that the PL condition holds globally or locally for deep linear residual network (Hardt & Ma, 2016), deep linear network, one hidden layer neural network with Leaky ReLU activation (Charles & Papailiopoulos, 2017; Zhou & Liang, 2017). Several studies (Li & Yuan, 2017; Arora et al., 2018; Allen-Zhu et al., 2018; Du et al., 2018b; Li & Liang, 2018) consider the trajectory of (stochastic) gradient descent on learning neural networks, and their analysis imply the PL condition in a certain form. For example, Du et al. (2018b) show that when

the width of a two layer neural network is sufficiently large, a global optimum would lie in the ball centered at the initial solution, in which PL condition holds. Allen-Zhu et al. (2018) extends this insight further to overparameterized deep neural networks with ReLU activation, and show that the PL condition holds for a global minimum around a random initial solution.

# 3 PRELIMINARIES AND NOTATIONS

Let $\|\cdot\|$ denote the Euclidean norm. A function $f(\mathbf{x})$ is $\rho$-weakly convex if $f(\mathbf{x}) + \frac{\rho}{2}\|\mathbf{x}\|^2$ is convex, where $\rho$ is the so-called weak-convexity parameter. A function $f(\mathbf{x})$ satisfies PL condition with parameter $\mu > 0$ if $f(\mathbf{x}) - f(\mathbf{x}_*) \leq \frac{1}{2\mu}\|\nabla f(\mathbf{x})\|^2$, where $\mathbf{x}_*$ stands for the optimal solution of $f$. Let $\mathbf{z} = (\mathbf{x}, y) \sim \mathbb{P}$ denote a random data following an unknown distribution $\mathbb{P}$, where $\mathbf{x} \in \mathcal{X}$ represents the feature vector and $y \in \mathcal{Y} = \{-1, +1\}$ represents the label. Denote by $\mathcal{Z} = \mathcal{X} \times \mathcal{Y}$ and by $p = \Pr(y = 1) = \mathbb{E}_y\left[\mathbb{I}_{[y=1]}\right]$, where $\mathbb{I}(\cdot)$ is the indicator function.

The area under the curve (AUC) on a population level for a scoring function $h : \mathcal{X} \rightarrow \mathbb{R}$ is defined as
$$\text{AUC}(h) = \Pr\left(h(\mathbf{x}) \geq h(\mathbf{x}')|y = 1, y' = -1\right),$$
where $\mathbf{z} = (\mathbf{x}, y)$ and $\mathbf{z}' = (\mathbf{x}', y')$ are drawn independently from $\mathbb{P}$. By employing the squared loss as the surrogate for the indicator function that is a common choice used by previous studies (Ying et al., 2016; Gao et al., 2013), the AUC maximization problem can be formulated as
$$\min_{h \in \mathcal{H}} \mathbb{E}_{\mathbf{z}, \mathbf{z}'}\left[(1 - h(\mathbf{x}) + h(\mathbf{x}'))^2|y = 1, y' = -1\right],$$
where $\mathcal{H}$ denotes a hypothesis class. All previous works of AUC maximization assume $h(\mathbf{x}) = \mathbf{w}^\top \mathbf{x}$ for simplicity. Instead, we consider learning a general nonlinear model parameterized by $\mathbf{w}$, i.e. $h(\mathbf{w}; \mathbf{x})$, which is not necessarily linear or convex in terms of $\mathbf{w}$ (e.g., $h(\mathbf{w}; \mathbf{x})$ can be a score function defined by a neural network with weights denoted by $\mathbf{w}$). Hence, the corresponding optimization problem becomes
$$\min_{\mathbf{w} \in \mathbb{R}^d} P(\mathbf{w}) := \mathbb{E}_{\mathbf{z}, \mathbf{z}'}\left[(1 - h(\mathbf{w}; \mathbf{x}) + h(\mathbf{w}; \mathbf{x}'))^2|y = 1, y' = -1\right] \tag{1}$$

The following proposition converts the original optimization problem (1) into a saddle-point problem, which is similar to Theorem 1 in (Ying et al., 2016). For completeness, the proof is included in the supplement.

**Proposition 1.** *The optimization problem (1) is equivalent to*
$$\min_{\mathbf{w} \in \mathbb{R}^d, (a,b) \in \mathbb{R}^2} \max_{\alpha \in \mathbb{R}} f(\mathbf{w}, a, b, \alpha) := \mathbb{E}_{\mathbf{z}}\left[F(\mathbf{w}, a, b, \alpha; \mathbf{z})\right], \tag{2}$$
*where $\mathbf{z} = (\mathbf{x}, y) \sim \mathbb{P}$, and*
$$F(\mathbf{w}, a, b, \alpha, \mathbf{z}) = (1 - p)(h(\mathbf{w}; \mathbf{x}) - a)^2 \mathbb{I}_{[y=1]} + p(h(\mathbf{w}; \mathbf{x}) - b)^2 \mathbb{I}_{[y=-1]}$$
$$+ 2(1 + \alpha)\left(ph(\mathbf{w}; \mathbf{x})\mathbb{I}_{[y=-1]} - (1-p)h(\mathbf{w}; \mathbf{x})\mathbb{I}_{[y=1]}\right) - p(1-p)\alpha^2$$

**Remark**: It is notable that the min-max formulation (2) is more favorable than the original formulation (1) for developing a stochastic algorithm that updates the model parameters based on one example or a mini-batch of samples. For stochastic optimization of (1), one has to carefully sample both positive and negative examples, which is not allowed in an online setting. It is notable that in the classical batch-learning setting, $p$ becomes the ratio of positive training examples and the expectation in (2) becomes average over $n$ individual functions. However, our algorithms are applicable to both batch-learning setting and online learning setting.

Define $\mathbf{v} = (\mathbf{w}^\top, a, b)^\top$, $\phi(\mathbf{v}) = \max_\alpha f(\mathbf{v}, \alpha)$. It is clear that $\min_{\mathbf{w}} P(\mathbf{w}) = \min_{\mathbf{v}} \phi(\mathbf{v})$ and $P(\mathbf{w}) \leq \phi(\mathbf{v})$ for any $\mathbf{v} = (\mathbf{w}^\top, a, b)^\top$. The following assumption is made throughout the paper.

**Assumption 1.** *(1) $\mu(\phi(\mathbf{v}) - \phi(\mathbf{v}_*)) \leq \frac{1}{2}\|\nabla\phi(\mathbf{v})\|^2$, where $\mu > 0$ and $\mathbf{v}_*$ is the optimal solution of $\phi$. (2) $h(\mathbf{w}; \mathbf{x})$ is $\tilde{L}$-Lipschitz continuous in terms of $\mathbf{w}$ for all $\mathbf{x}$. (3) $\phi(\mathbf{v})$ is $L$-smooth. (4) Var$[h(\mathbf{w}; \mathbf{x})|y = -1] \leq \sigma^2$, Var$[h(\mathbf{w}; \mathbf{x})|y = 1] \leq \sigma^2$. (5) $0 \leq h(\mathbf{w}; \mathbf{x}) \leq 1$. (6) Given a initial solution $\bar{\mathbf{v}}_0$, there exists $\Delta_0 > 0$ such that $\phi(\bar{\mathbf{v}}_0) - \phi(\mathbf{v}_*) \leq \Delta_0$, where $\mathbf{v}_*$ is the global minimum of $\phi$.*

**Remark**: The first condition is inspired by a PL condition on the objective function $P(\mathbf{w})$ for learning a deep neural network. and the following Lemma 1 establishes the connection. $h(\mathbf{w}; \mathbf{x}) \in [0, 1]$ holds when $h$ is defined as the sigmoid function composited with the forward propagation function of a neural network.

---

**Algorithm 1** Proximally Guided Algorithm (PGA) (Rafique et al., 2018)

---
1: Initialize $\bar{\mathbf{v}}_0 = \mathbf{0} \in \mathbb{R}^{d+2}$, $\bar{\alpha}_0 = 0$, the global index $j = 0$
2: **for** $k = 1, \ldots, K$ **do**
3:    $\mathbf{v}_0^k = \bar{\mathbf{v}}_{k-1}$, $\alpha_0^k = \bar{\alpha}_{k-1}$, $\eta_k = \eta_0/k$, $T_k = T_0 \cdot k^2$
4:    **for** $t = 1, \ldots, T_k$ **do**
5:       Receive $\mathbf{z}_j = (\mathbf{x}_j, y_j)$ from $\mathbb{P}$, $\hat{\mathbf{g}}_{\mathbf{v}} = \nabla_{\mathbf{v}} F(\mathbf{v}_{t-1}^k, \alpha_{t-1}^k; \mathbf{z}_j)$, $\hat{\mathbf{g}}_{\alpha} = \nabla_{\alpha} F(\mathbf{v}_{t-1}^k, \alpha_{t-1}^k; \mathbf{z}_j)$
6:       $\mathbf{v}_t^k = \Pi_{\Omega_1} \left[ \mathbf{v}_{t-1}^k - \eta_k \left( \hat{\mathbf{g}}_{\mathbf{v}} + \frac{1}{\gamma}(\mathbf{v}_{t-1}^k - \mathbf{v}_0^k) \right) \right]$, where $\Omega_1 = \{\mathbf{v} : \|\mathbf{v}\| \leq R_1\}$
7:       $\alpha_t^k = \Pi_{\Omega_2} \left[ \alpha_{t-1}^k + \eta_k \hat{\mathbf{g}}_{\alpha} \right]$, where $\Omega_2 = \{\alpha : |\alpha| \leq R_2\}$
8:    **end for**
9:    $\bar{\mathbf{v}}_k = \frac{1}{T_k} \sum_{t=1}^{T_k} \mathbf{v}_t^k$, $\bar{\alpha}_k = \frac{1}{T_k} \sum_{t=1}^{T_k} \alpha_t^k$
10: **end for**
11: Sample $\tau$ uniformly randomly from $\{1, \ldots, K\}$
12: **return** $\bar{\mathbf{v}}_\tau, \bar{\alpha}_\tau$

---

**Lemma 1.** *Suppose $\|\nabla_{\mathbf{w}} h(\mathbf{w}; \mathbf{x})\| \leq \tilde{L}$ for all $\mathbf{w}$ and $\mathbf{x}$. If $P(\mathbf{w})$ satisfies PL condition, i.e. there exists $\mu' > 0$, such that $\mu'(P(\mathbf{w}) - \min_{\mathbf{w}} P(\mathbf{w})) \leq \frac{1}{2} \|\nabla_{\mathbf{w}} P(\mathbf{w})\|^2$, then we have $\mu(\phi(\mathbf{v}) - \phi(\mathbf{v}_*)) \leq \frac{1}{2} \|\nabla\phi(\mathbf{v})\|^2$, where $\mu = \frac{1}{\max\left( \frac{1}{2\min(p, 1-p)} + \frac{2\tilde{L}^2}{\mu' \min(p^2, (1-p)^2)}, \frac{2}{\mu'} \right)}$.*

**Remark**: The PL condition of $P(\mathbf{w})$ could be proved for learning a neural network similar to existing studies, which is not the main focus of this paper. Nevertheless, In Appendix A.7, we provide an example for AUC maximization with one-hidden layer neural network.

**Warmup.** We first discuss the algorithms and their convergence results of (Rafique et al., 2018) applied to the considered min-max problem. They have algorithms for problems in batch-learning setting and online learning setting. Since the algorithms for the batch-learning setting have complexities scaling with $n$, we will concentrate on the algorithm for the online learning setting. The algorithm is presented in Algorithm 1, which is a direct application of Algorithm 2 of (Rafique et al., 2018) to an online setting. Since their analysis requires the domain of the primal and the dual variable to be bounded, hence we add a ball constraint on the primal variable and the dual variable as well. As long as $R_1$ and $R_2$ is sufficiently large, they should not affect the solution. The convergence result of Algorithm 1 is stated below.

**Theorem 1.** *(Rafique et al., 2018) Suppose $f(\mathbf{v}, \alpha)$ is $\rho$-weakly convex in $\mathbf{v}$ and concave in $\alpha$. Let $\gamma = 1/2\rho$, and define $\hat{\mathbf{v}}_\tau = \arg\min_{\mathbf{v}} \phi(\mathbf{v}) + \frac{1}{2\gamma} \|\mathbf{v} - \bar{\mathbf{v}}_\tau\|^2$. Algorithm 1 with $T_k = ck^2$ and $K = \widetilde{O}(\epsilon^{-2})$ ensures that $\mathbb{E}\left[ dist^2(0, \partial\phi(\hat{\mathbf{v}}_\tau)) \right] \leq \frac{1}{\gamma^2} \mathbb{E}\|\hat{\mathbf{v}}_\tau - \bar{\mathbf{v}}_\tau\|^2 \leq \epsilon^2$. The total iteration complexity is $\widetilde{O}(\epsilon^{-6})$.*

**Remark:** Under the condition $\phi(\mathbf{v})$ is smooth and the returned solution is within the added bounded ball constraint, the above result implies $\mathbb{E}[\|\nabla\phi(\bar{\mathbf{v}}_\tau)\|^2 \leq \epsilon]$ with a complexity of $\widetilde{O}(1/\epsilon^3)$. It further implies that with a complexity of $\widetilde{O}(1/(\mu^3\epsilon^3))$ we have $\mathbb{E}[\phi(\bar{\mathbf{v}}_\tau) - \min_{\mathbf{v}} \phi(\mathbf{v})] \leq \epsilon$ under the assumed PL condition.

We can see that this complexity result under the PL condition of $\phi(\mathbf{v})$ is worse than the typical complexity result of stochastic gradient descent method under the PL condition (i.e., $O(1/\epsilon)$) (Karimi et al., 2016). It remains an open problem how to design a stochastic primal-dual algorithm for solving $\min_{\mathbf{v}} \max_{\alpha} F(\mathbf{v}, \alpha)$ in order to achieve a complexity of $O(1/\epsilon)$ in terms of minimizing $\phi(\mathbf{v})$. A naive idea is to solve the inner maximization problem of $\alpha$ first and the use SGD on the primal variable $\mathbf{v}$. However, this is not viable since exact maximization over $\alpha$ is a non-trivial task.

## 4 ALGORITHMS AND THEORETICAL ANALYSIS

In this section, we present two primal-dual algorithms for solving the min-max optimization problem (2) with corresponding theoretical convergence results. For simplicity, we first assume the positive ratio $p$ is known in advance, which is true in the batch-learning setting. Handling the unknown $p$ in an online learning setting is a simple extension, which will be discussed in Section 4.3. The proposed algorithms follow the same proximal point framework proposed in (Rafique et al., 2018), i.e., we

---

**Algorithm 2** Proximal Primal-Dual Stochastic Gradient (PPD-SG)

---

1: Initialize $\bar{\mathbf{v}}_0 = \mathbf{0} \in \mathbb{R}^{d+2}$, $\bar{\alpha}_0 = 0$, the global index $j = 0$
2: **for** $k = 1, \ldots, K$ **do**
3:     $\mathbf{v}_0^k = \bar{\mathbf{v}}_{k-1}$, $\alpha_0^k = \bar{\alpha}_{k-1}$, $\eta_k = \eta_0 \exp\left(-(k-1)\frac{\mu/L}{5+\mu/L}\right)$
4:     **for** $t = 1, \ldots, T_k - 1$ **do**
5:         Receive $\mathbf{z}_j = (\mathbf{x}_j, y_j)$ from $\mathbb{P}$, $\hat{\mathbf{g}}_{\mathbf{v}} = \nabla_{\mathbf{v}} F(\mathbf{v}_{t-1}^k, \alpha_{t-1}^k; \mathbf{z}_j)$, $\hat{\mathbf{g}}_\alpha = \nabla_\alpha F(\mathbf{v}_{t-1}^k, \alpha_{t-1}^k; \mathbf{z}_j)$
6:         $\mathbf{v}_t^k = \mathbf{v}_{t-1}^k - \eta_k\left(\hat{\mathbf{g}}_{\mathbf{v}} + \frac{1}{\gamma}(\mathbf{v}_{t-1}^k - \mathbf{v}_0^k)\right)$
7:         $\alpha_t^k = \alpha_{t-1}^k + \eta_k\hat{\mathbf{g}}_\alpha$
8:         $j = j + 1$
9:     **end for**
10:    $\bar{\mathbf{v}}_k = \frac{1}{T_k}\sum_{t=0}^{T_k-1} \mathbf{v}_t^k$
11:    Draw a minibatch $\{\mathbf{z}_j, \ldots, \mathbf{z}_{j+m_k-1}\}$ of size $m_k$
12:    $\bar{\alpha}_k = \frac{\sum_{i=j}^{j+m_k-1} h(\bar{\mathbf{w}}_k; \mathbf{x}_i)\mathbb{I}_{y_i=-1}}{\sum_{i=j}^{j+m_k-1}\mathbb{I}_{y_i=-1}} - \frac{\sum_{i=j}^{j+m_k-1} h(\bar{\mathbf{w}}_k; \mathbf{x}_i)\mathbb{I}_{y_i=1}}{\sum_{i=j}^{j+m_k-1}\mathbb{I}_{y_i=1}}$
13:    $j = j + m_k$
14: **end for**
15: **return** $\bar{\mathbf{v}}_K, \bar{\alpha}_K$

---

solve the following convex-concave problems approximately and iteratively:

$$\min_{\mathbf{v}} \max_{\alpha \in \mathbb{R}} \{f(\mathbf{v}, \alpha) + \frac{1}{2\gamma}\|\mathbf{v} - \mathbf{v}_0\|^2\} \tag{3}$$

where $\gamma < 1/L$ to ensure that the new objective function becomes convex and concave, and $\mathbf{v}_0$ is periodically updated.

### 4.1 PROXIMAL PRIMAL-DUAL STOCHASTIC GRADIENT

Our first algorithm named Proximal Primal-Dual Stochastic Gradient (PPD-SG) is presented in Algorithm 2. Similar to Algorithm 1, it has a nested loop, where the inner loop is to approximately solve a regularized min-max optimization problem (3) using stochastic primal-dual gradient method, and the outer loop updates the reference point and learning rate. One key difference is that PPD-SG uses a geometrically decaying step size scheme, while Algorithm 1 uses a polynomially decaying step size scheme. Another key difference is that at the end of $k$-th outer loop, we update the dual variable $\bar{\alpha}_k$ in Step 12, which is motivated by its closed-form solution given $\bar{\mathbf{v}}_k$. In particular, the given $\bar{\mathbf{v}}_k$, the dual solution that optimizes the inner maximization problem is given by:

$$\alpha = \frac{\mathbb{E}[h(\bar{\mathbf{w}}_k; \mathbf{x})I_{y=-1}]}{1-p} - \frac{\mathbb{E}[h(\bar{\mathbf{w}}_k; \mathbf{x})I_{y=1}]}{p} = \mathbb{E}_{\mathbf{x}}[h(\bar{\mathbf{w}}_k; \mathbf{x})|y = -1] - \mathbb{E}_{\mathbf{x}}[h(\bar{\mathbf{w}}_k; \mathbf{x})|y = 1].$$

In the algorithm, we only use a small number of samples in Step 11 to compute an estimation of the optimal $\alpha$ given $\bar{\mathbf{v}}_k$. These differences are important for us to achieve lower iteration complexity of PPD-SG. Next, we present our convergence results of PPD-SG.

**Lemma 2** (One Epoch Analysis of Algorithm 2). *Suppose Assumption 1 and there exists $G > 0$ such that $\|\hat{\mathbf{g}}_t^k\|_2 \leq G$, where $\hat{\mathbf{g}}_t^k = \left(\nabla_{\mathbf{v}} F(\mathbf{v}_t^k, \alpha_t^k; \mathbf{z})^\top + \frac{1}{\gamma}\left(\mathbf{v}_t^k - \mathbf{v}_0^k\right)^\top, -\nabla_\alpha F(\mathbf{v}_t^k, \alpha_t^k; \mathbf{z})^\top\right)^\top$. Define $\phi_k(\mathbf{v}) = \phi(\mathbf{v}) + \frac{1}{2\gamma}\|\mathbf{v} - \bar{\mathbf{v}}_{k-1}\|^2$, $\mathbf{s}_k = \arg\min_{\mathbf{v} \in \mathbb{R}^{d+2}} \phi_k(\mathbf{v})$. Choosing $m_{k-1} \geq \frac{2(\sigma^2+C)}{p(1-p)\eta_k^2 G^2 T_k}$ with $C = \frac{2}{\ln(\frac{1}{\max(p,1-p)})}\max(p, 1-p)^{\frac{1}{\ln(1/\max(p,1-p))}}$, then we have*

$$\mathbb{E}_{k-1}\left[\phi_k(\bar{\mathbf{v}}_k) - \min_{\mathbf{v}} \phi_k(\mathbf{v})\right] \leq \frac{\|\bar{\mathbf{v}}_{k-1} - \mathbf{s}_k\|^2 + 16\tilde{L}^2\mathbb{E}_{k-1}\|\bar{\mathbf{v}}_{k-1} - \bar{\mathbf{v}}_k\|^2}{2\eta_k T_k} + 4\eta_k G^2.$$

*where $\mathbb{E}_{k-1}$ stands for the conditional expectation conditioning on all the stochastic events until $\bar{\mathbf{v}}_{k-1}$ is generated.*

**Theorem 2.** *Suppose the same conditions in Lemma 2 hold. Set $\eta_k = \eta_0 \exp\left(-(k-1)\frac{\mu/L}{5+\mu/L}\right)$ and $T_k = \frac{\max(2, 16\tilde{L}^2)}{L\eta_0}\exp\left((k-1)\frac{\mu/L}{5+\mu/L}\right)$, $m_k = \frac{2(\sigma^2+C)L}{p(1-p)G^2\eta_0 \max(2,16\tilde{L}^2)}\exp\left(k\frac{\mu/L}{5+\mu/L}\right)$ with $C = \frac{2}{\ln(\frac{1}{\max(p,1-p)})}\max(p, 1-p)^{\frac{1}{\ln(1/\max(p,1-p))}}$, $\gamma = \frac{1}{2L}$ in Algorithm 2. To return $\bar{\mathbf{v}}_K$ such that $\mathbb{E}[\phi(\bar{\mathbf{v}}_K) - \phi(\mathbf{v}_*)] \leq \epsilon$, it suffices to choose $K \geq \left(\frac{5L}{\mu} + 1\right)\max\left(\log\frac{2\Delta_0}{\epsilon}, \log K + \log\frac{48G^2\eta_0}{5\epsilon}\right).$*

---

**Algorithm 3** Inner Loop of Proximal Primal-Dual AdaGrad (PPD-AdaGrad)

1: **for** $t = 1, \ldots, T_k - 1$ **do**
2:     Receive $\mathbf{z}_j = (\mathbf{x}_j, y_j)$ from $\mathbb{P}$, $\hat{\mathbf{g}}_{\mathbf{v}} = \nabla_{\mathbf{v}} F(\mathbf{v}_t^k, \alpha_t^k; \mathbf{z}_j)$, $\hat{\mathbf{g}}_\alpha = \nabla_\alpha F(\mathbf{v}_t^k, \alpha_t^k; \mathbf{z}_j)$
3:     $\hat{\mathbf{g}}_t^k = [\hat{\mathbf{g}}_{\mathbf{v}} + \frac{1}{\gamma}(\mathbf{v}_t^k - \mathbf{v}_0^k); -\hat{\mathbf{g}}_\alpha] \in \mathbb{R}^{d+3}$, $\hat{\mathbf{g}}_{1:t}^k = [\hat{\mathbf{g}}_{1:t-1}^k, \hat{\mathbf{g}}_t^k]$, $s_{t,i}^k = \|\hat{\mathbf{g}}_{1:t,i}^k\|_2$,
4:     $H_t^k = \delta I + \text{diag}(s_t^k)$, $\psi_t^k(\mathbf{u}) = \frac{1}{2}\langle \mathbf{u} - \mathbf{u}_0^k, H_t^k(\mathbf{u} - \mathbf{u}_0^k) \rangle$, where $\mathbf{u}_0^k = [\mathbf{v}_0^k; \alpha_0^k] \in \mathbb{R}^{d+3}$
5:     $\mathbf{u}_{t+1}^k = \arg\min_{\mathbf{u}} \left\{ \eta_k \langle \frac{1}{t}\sum_{\tau=1}^t \hat{\mathbf{g}}_\tau^k, \mathbf{u}\rangle + \frac{1}{t}\psi_t^k(\mathbf{u}) \right\}$
6: **end for**

---

*The number of iterations is at most* $\widetilde{O}\left(\frac{LG^2}{\mu^2\epsilon}\right)$, *and the required number of samples is at most* $\widetilde{O}\left(\frac{L^3\sigma^2}{\mu^2\epsilon}\right)$, *where* $\widetilde{O}(\cdot)$ *hides logarithmic factors of* $L, \mu, \epsilon, \delta$. *where* $\widetilde{O}(\cdot)$ *hides logarithmic factor of* $L, \mu, \epsilon, G, \sigma$.

**Remark**: The above complexity result is similar to that of (Karimi et al., 2016) for solving non-convex minimization problem under the PL condition up to a logarithmic factor. Compared with the complexity result of Algorithm 1 discussed earlier, i.e., $\widetilde{O}(1/(\mu^3\epsilon^3))$, the above complexity in the order of $\widetilde{O}(1/(\mu^2\epsilon))$ is much better - it not only improves the dependence on $\epsilon$ but also improves the dependence on $\mu$.

## 4.2 PROXIMAL PRIMAL-DUAL ADAGRAD

Our second algorithm named Proximal Primal-Dual Adagrad (PPD-Adagrad) is a AdaGrad-style algorithm. Since it only differs from PPD-SG in the updates of the inner loop, we only present the inner loop in Algorithm 3. The updates in the inner loop are similar to the adaptive updates of traditional AdaGrad (Duchi et al., 2011). We aim to achieve an adaptive convergence by using PPD-AdaGrad. The analysis of PPD-AdaGrad is inspired by the analysis of AdaGrad for non-convex minimization problems (Chen et al., 2019). The key difference is that we have to carefully deal with the primal-dual updates for the non-convex min-max problem. We summarize the convergence results of PPD-AdaGrad below.

**Lemma 3** (One Epoch Analysis of Algorithm 3). *Suppose Assumption 1 and* $\|\hat{\mathbf{g}}_t^k\|_\infty \leq \delta$ *hold. Define* $\phi_k(\mathbf{v}) = \phi(\mathbf{v}) + \frac{1}{2\gamma}\|\mathbf{v} - \bar{\mathbf{v}}_{k-1}\|^2$, $\mathbf{s}_k = \arg\min_{\mathbf{v}\in\mathbb{R}^{d+2}} \phi_k(\mathbf{v})$. *Choosing* $m_{k-1} \geq \frac{2(\sigma^2+C)}{p(1-p)(d+3)\eta_k^2}$ *with* $C = \frac{2}{\ln(\frac{1}{\max(p,1-p)})}\max(p, 1-p)^{\frac{1}{\ln(1/\max(p,1-p))}}$, *and*

$$T_k = \inf\left\{\tau : \tau \geq M_k \max\left(\frac{(\delta + \max_i \|\hat{\mathbf{g}}_{1:\tau,i}^k\|_2)\max(1, 8\tilde{L}^2)}{c}, 2c(\sum_{i=1}^{d+3}\|\hat{\mathbf{g}}_{1:\tau,i}^k\|_2 + (d+3)(\delta + \max_i \|\hat{\mathbf{g}}_{1:\tau,i}^k\|_2))\right)\right\}$$

*with* $M_k > 0, c > 0$, *then we have*

$$\mathbb{E}_{k-1}\left[\phi_k(\bar{\mathbf{v}}_k) - \min_{\mathbf{v}}\phi_k(\mathbf{v})\right] \leq \frac{c\left(\|\bar{\mathbf{v}}_{k-1} - \mathbf{s}_k\|_2^2 + \mathbb{E}_{k-1}\|\bar{\mathbf{v}}_{k-1} - \bar{\mathbf{v}}_k\|_2^2\right)}{\eta_k M_k} + \frac{\eta_k}{cM_k}.$$

*where* $\mathbb{E}_{k-1}$ *stands for the conditional expectation conditioning on all the stochastic events until* $\bar{\mathbf{v}}_{k-1}$ *is generated.*

**Theorem 3.** *Suppose the same conditions as in Lemma 3 hold. Set* $\eta_k = \eta_0\exp\left(-\frac{(k-1)}{2}\frac{\mu/L}{5+\mu/L}\right)$, $M_k = \frac{4c}{L\eta_0}\exp\left(\frac{(k-1)}{2}\frac{\mu/L}{5+\mu/L}\right)$, $m_k = \frac{2(\sigma^2+C)}{p(1-p)\eta_0^2(d+3)}\exp\left(k\frac{\mu/L}{5+\mu/L}\right)$ *with* $C = \frac{2}{\ln(\frac{1}{\max(p,1-p)})}\max(p, 1-p)^{\frac{1}{\ln(1/\max(p,1-p))}}$, $\gamma = \frac{1}{2L}$ *and* $T_k$ *as in Lemma 3 where* $c = \frac{1}{\sqrt{d+3}}$. *Suppose* $\|\hat{\mathbf{g}}_{1:T_k,i}^k\|_2 \leq \delta \cdot T_k^\alpha$ *for* $\forall k$, *where* $0 \leq \alpha \leq \frac{1}{2}$. *To return* $\bar{\mathbf{v}}_K$ *such that* $\mathbb{E}\left[\phi(\bar{\mathbf{v}}_K) - \phi(\mathbf{v}_*)\right] \leq \epsilon$, *it suffices to choose* $K \geq \left(\frac{5L}{\mu} + 1\right)\max\left(\log\frac{2\Delta_0}{\epsilon}, \log K + \log\frac{\eta_0^2 L}{5c^2\epsilon}\right)$. *The number of iterations is at most* $\widetilde{O}\left(\left(\frac{L\delta^2 d}{\mu^2\epsilon}\right)^{\frac{1}{2(1-\alpha)}}\right)$, *and the required number of samples is at most* $\widetilde{O}\left(\frac{L^3\sigma^2}{\mu^2\epsilon}\right)$, *where* $\widetilde{O}(\cdot)$ *hides logarithmic factors of* $L, \mu, \epsilon, \delta$.

**Remark**: When the cumulative growth of stochastic gradient is slow, i.e., $\alpha < 1/2$, the number of iterations is less than that in Theorem 2, which exhibits adaptive iteration complexity.

## 4.3 EXTENSIONS

---

**Algorithm 4** Update $T_+, T_-, \widehat{p}, \widehat{p(1-p)}, \bar{y}$ given data $\{\mathbf{z}_j, \ldots, \mathbf{z}_{j+m-1}\}$

---

1: Update $T_- = T_- + \sum_{i=j}^{j+m-1} \mathbb{I}_{[y_j = -1]}, T_+ = T_+ + \sum_{i=j}^{j+m-1} \mathbb{I}_{[y_j = 1]}$

2: $\widehat{p} = T_+/(T_+ + T_-), \bar{y} = \frac{(j+2)\bar{y} + \sum_{i=j}^{j+m-1} \mathbb{I}_{[y_i=1]}}{j+m+2}, \widehat{p(1-p)} = \frac{(j+1)\widehat{p(1-p)} + \sum_{i=j}^{j+m-1}(\mathbb{I}_{[y_i=1]} - \bar{y})^2}{j+m+1}$

---

**Setting** $\eta_k, T_k, m_k$. It is notable that the setting of $\eta_k, T_k, m_k$ depends on unknown parameters $\mu$, $L$, etc., which are typically unknown. One heuristic to address this issue is that we can decrease $\eta_k$ by a constant factor larger than 1 (e.g., 2 or 5 or 10), and similarly increase $T_k$ and $m_k$ by a constant factor. Another heuristic is to decrease the step size by a constant factor when the performance on a validation data saturates (Krizhevsky et al., 2012).

**Variants when $p$ is unknown**. In the online learning setting when $p$ is unknown, the stochastic gradients of $f$ in both $\mathbf{v}$ and $\alpha$ are not directly available. To address this issue, we can keep unbiased estimators for both $p$ and $p(1-p)$ which are independent of the new arrived data, and update these estimators during the optimization procedure. All values depending on $p$ and $p(1-p)$ (i.e., $F, \mathbf{g_v}, \mathbf{g}_\alpha$) are estimated by substituting $p$ and $p(1-p)$ by $\widehat{p}$ and $\widehat{p(1-p)}$ (i.e., $\hat{F}, \hat{\mathbf{g}}_\mathbf{v}, \hat{\mathbf{g}}_\alpha$) respectively. The approach for keeping unbiased estimator $\hat{p}$ and $\widehat{p(1-p)}$ during the optimization is described in Algorithm 4, where $j$ is the global index, and $m$ is the number of examples received.

**Extensions to multi-class problems.** In the previous analysis, we only consider the binary classification problem. We can extend it to the multi-class setting. To this end, we first introduce the definition of AUC in this setting according to (Hand & Till, 2001). Suppose there are $c$ classes, we have $c$ scoring functions for each class, namely $h(\mathbf{w}_1; \mathbf{x}), \ldots, h(\mathbf{w}_c; \mathbf{x})$. We assume that these scores are normalized such that $\sum_{k=1}^{c} h(\mathbf{w}_c; \mathbf{x}) = 1$. Note that if these functions are implemented by a deep neural network, they can share the lower layers and have individual last layer of connections. The AUC is defined as

$$\text{AUC}(h) = \frac{1}{c(c-1)} \sum_{i=1}^{c} \sum_{j \neq i} \Pr\left(h(\mathbf{w}_i, \mathbf{x}) \geq h(\mathbf{w}_i; \mathbf{x}') | y = i, y' = j\right), \quad (4)$$

Similar to Proposition 1, we can cast the problem into

$$\min_{\mathbf{w}, \mathbf{a}, \mathbf{b}} \max_{\alpha} \frac{1}{c(c-1)} \sum_{i=1}^{c} \sum_{j \neq i} \mathbb{E}_\mathbf{z} \left[F_{ij}\left(\mathbf{w}_i, a_{ij}, b_{ij}, \alpha_{ij}; \mathbf{z}\right)\right], \quad (5)$$

where $\mathbf{w} = [\mathbf{w}_1, \ldots, \mathbf{w}_c], \mathbf{a}, \mathbf{b}, \alpha \in \mathbb{R}^{c \times c}, i, j = 1, \ldots, c, \mathbf{z} = (\mathbf{x}, y) \sim \mathbb{P}, p_i = \Pr(y = i)$, and

$$F_{ij}(\mathbf{w}_i, a_{ij}, b_{ij}, \alpha_{ij}, \mathbf{z}) = p_j \left(h(\mathbf{w}_i; \mathbf{x}) - a_{ij}\right)^2 \mathbb{I}_{[y=i]} + p_i (h(\mathbf{w}_i; \mathbf{x}) - b_{ij})^2 \mathbb{I}_{[y=j]}$$
$$+ 2(1 + \alpha_{ij})\left(p_i h(\mathbf{w}_i; \mathbf{x})\mathbb{I}_{[y=j]} - p_j h(\mathbf{w}_i; \mathbf{x})\mathbb{I}_{[y=i]}\right) - p_i p_j \alpha_{ij}^2.$$

Then we can modify our algorithms to accommodate the multiple class pairs. We can also add another level of sampling of class pairs into computing the stochastic gradients.

## 5 EXPERIMENTAL RESULTS

In this section, we present some empirical results to verify the effectiveness of the proposed algorithms. We compare our algorithms (PPD-SG and PPD-AdaGrad) with three baseline methods including PGA (Algorithm 1), Online AUC method (Ying et al., 2016) (OAUC) that directly employs the standard primal-dual stochastic gradient method with a decreasing step size for solving the min-max formulation, and the standard stochastic gradient descent (SGD) for minimizing cross-entropy loss. Comparing with PGA and OAUC allows us to verify the effectiveness of the proposed algorithms for solving the same formulation, and comparing with SGD allows us to verify the effectiveness of maximizing AUC for imbalanced data. We use a residual network with 20 layers (ResNet-20) to implement the deep neural network for all algorithms.

We use the stagewise step size strategy as in (He et al., 2016) for SGD, i.e. the step size is decreased by 10 times at 40K, 60K. For PPD-SG and PPD-AdaGrad, we set $T_s = T_0 3^k, \eta_s = \eta_0/3^k$. $T_0, \eta_0$ are tuned on a validation data. The value of $\gamma$ is tuned for PGA and the same value is used for PPD-SG and PPD-AdaGrad. The initial step size is tuned in [0.1, 0.05, 0.01, 0.008, 0.005] and $T_0$ is tuned in $[200 \sim 2000]$ for each algorithm separately. The batch size is set to 128. For STL10, we use a smaller batch size 32 due to the limited training data.

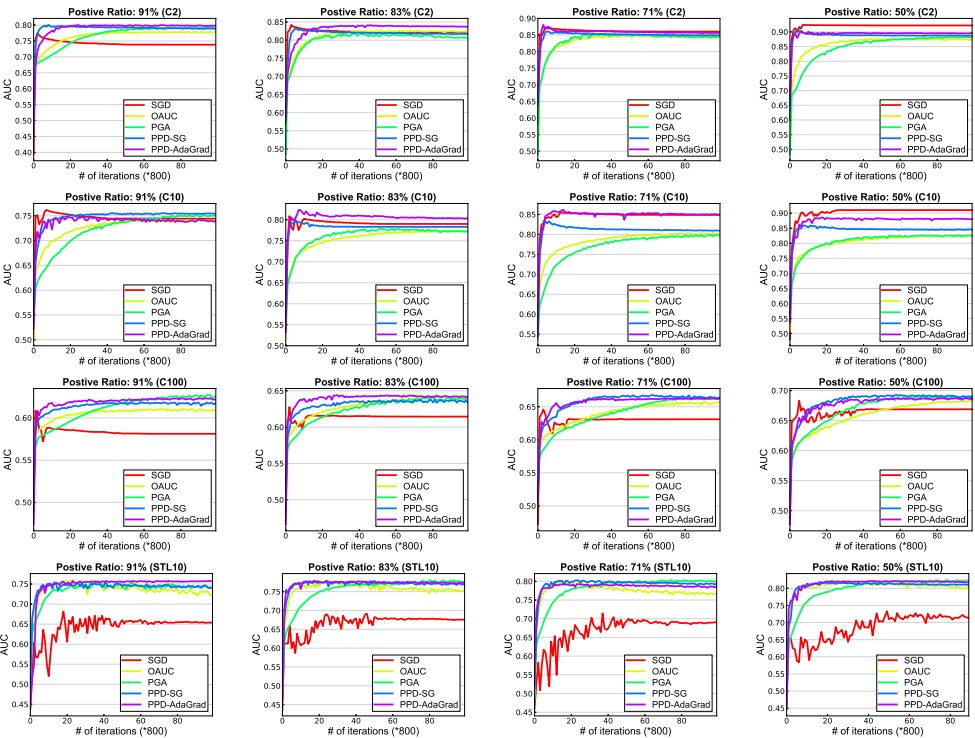

Figure 1: Comparison of testing AUC on Cat&Dog, CIFAR10, CIFAR100 and STL10.

We conduct the comparisons on four benchmark datasets, i.e., Cat&Dog (C2), CIFAR10 (C10), CIFAR100 (C100), STL10. STL10 is an extension of CIFAR10 and the images are acquired from ImageNet. Cat&Dog is from Kaggle containing 25,000 images of dogs and cats and we choose an 80:20 split to construct training and testing set. We use 19k/1k, 45k/5k, 45k/5k, 4k/1k training/validation split on C2, C10, C100, and STL10 respectively. For each dataset, we construct multiple binary classification tasks with varying imbalanced ratio of number negative examples to number of positive examples. For details of construction of binary classification tasks, please refer to the Appendix A.8.

We report the convergence of AUC on testing data in Figure 1, where the title shows the ratio of the majority class to the minority class. The results about the convergence of AUC versus the time in seconds are also presented in Figure 3. From the results we can see that for the balanced settings with ratio equal to 50%, SGD performs consistently better than other methods on C2 and CIFAR10 data. However, it is worse than AUC optimization based methods on CIFAR100 and STL10. For imbalanced settings, AUC maximization based methods are more advantageous than SGD in most cases. In addition, PPD-SG and PPD-AdaGrad are mostly better than other baseline algorithms. In certain cases, PPD-AdaGrad can be faster than PPD-SG. Finally, we observe even better performance (in Appendix) by a mixed strategy that pre-trains the model with SGD and then switchs to PPD-SG.

# 6 CONCLUSION

In this paper, we consider stochastic AUC maximization problem when the predictive model is a deep neural network. By building on the saddle point reformulation and exploring Polyak-Łojasiewicz condition in deep learning, we have proposed two algorithms with state-of-the-art complexities for stochastic AUC maximization problem. We have also demonstrated the efficiency of our proposed algorithms on several benchmark datasets, and the experimental results indicate that our algorithms converge faster than other baselines. One may consider to extend the analysis techniques to other problems with the min-max formulation.

ACKNOWLEDGMENTS

The authors thank the anonymous reviewers for their helpful comments. M. Liu, Z. Yuan and T. Yang are partially supported by National Science Foundation CAREER Award 1844403.

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

# A  APPENDIX

## A.1  PROOF OF PROPOSITION 1

*Proof.* It suffices to prove that

$$
\mathbb{E}_{\mathbf{z},\mathbf{z}'} \left[ (1 - h(\mathbf{w};\mathbf{x}) + h(\mathbf{w};\mathbf{x}'))^2 \big| y = 1, y' = -1 \right] = 1 + \frac{\min_{(a,b)\in\mathbb{R}^2} \max_{\alpha\in\mathbb{R}} \mathbb{E}_{\mathbf{z}} \left[ F(\mathbf{w},a,b,\alpha;\mathbf{z}) \right]}{p(1-p)}
\tag{6}
$$

Note that

$$
\begin{aligned}
\text{LHS} &= 1 + \mathbb{E}\left[h^2(\mathbf{w};\mathbf{x})\big|y=1\right] + \mathbb{E}\left[h^2(\mathbf{w};\mathbf{x}')\big|y'=-1\right] - 2\mathbb{E}\left[h(\mathbf{w};\mathbf{x})|y=1\right] + 2\mathbb{E}\left[h(\mathbf{w};\mathbf{x}')|y'=-1\right] \\
&\quad - 2\left(\mathbb{E}\left[h(\mathbf{w};\mathbf{x})|y=1\right]\right)\left(\mathbb{E}\left[h(\mathbf{w};\mathbf{x}')|y'=-1\right]\right) \\
&= 1 + \mathbb{E}\left[h^2(\mathbf{w};\mathbf{x})\big|y=1\right] - \left(\mathbb{E}\left[h(\mathbf{w};\mathbf{x})|y=1\right]\right)^2 + \mathbb{E}\left[h^2(\mathbf{w};\mathbf{x}')\big|y'=-1\right] - \left(\mathbb{E}\left[h(\mathbf{w};\mathbf{x}')|y'=-1\right]\right)^2 \\
&\quad - 2\mathbb{E}\left[h(\mathbf{w};\mathbf{x})|y=1\right] + 2\mathbb{E}\left[h(\mathbf{w};\mathbf{x}')|y'=-1\right] + \left(\mathbb{E}[h(\mathbf{w};\mathbf{x})|y=1] - \mathbb{E}[h(\mathbf{w};\mathbf{x}')|y'=-1]\right)^2 \\
&= 1 + \min_{(a,b)\in\mathbb{R}^2} \mathbb{E}\left[(h(\mathbf{w};\mathbf{x}) - a)^2\big|y=1\right] + \mathbb{E}\left[(h(\mathbf{w};\mathbf{x}') - b)^2\big|y'=-1\right] - 2\mathbb{E}\left[h(\mathbf{w};\mathbf{x})|y=1\right] \\
&\quad + 2\mathbb{E}\left[h(\mathbf{w};\mathbf{x}')|y'=-1\right] + \max_{\alpha\in\mathbb{R}}\left[2\alpha\left(\mathbb{E}\left[h(\mathbf{w};\mathbf{x}')|y'=-1\right] - \mathbb{E}\left[h(\mathbf{w};\mathbf{x})|y=1\right]\right) - \alpha^2\right] \\
&= 1 + \min_{(a,b)\in\mathbb{R}^2} \max_{\alpha\in\mathbb{R}} \mathbb{E}_{\mathbf{z}} \left\{ \frac{1}{p}\left(h(\mathbf{w};\mathbf{x}) - a\right)^2 \mathbb{I}_{[y=1]} + \frac{1}{1-p}(h(\mathbf{w};\mathbf{x}) - b)^2 \mathbb{I}_{[y=-1]} \right. \\
&\quad \left. + 2(1+\alpha)\left(\frac{1}{1-p}h(\mathbf{w};\mathbf{x})\mathbb{I}_{[y=-1]} - \frac{1}{p}h(\mathbf{w};\mathbf{x})\mathbb{I}_{[y=1]}\right) - \alpha^2 \right\} \\
&= 1 + \frac{\min_{(a,b)\in\mathbb{R}^2} \max_{\alpha\in\mathbb{R}} \mathbb{E}_{\mathbf{z}}\left[F(\mathbf{w},a,b,\alpha;\mathbf{z})\right]}{p(1-p)} = \text{RHS}.
\end{aligned}
\tag{7}
$$

Note that the optimal values of $a, b, \alpha$ are chosen as $a^* = \mathbb{E}\left[h(\mathbf{w}; \mathbf{x}) | y = 1\right]$, $b = \mathbb{E}\left[h(\mathbf{w}; \mathbf{x}') | y' = -1\right]$, $\alpha^* = \mathbb{E}\left[h(\mathbf{w}; \mathbf{x}') | y' = -1\right] - \mathbb{E}\left[h(\mathbf{w}; \mathbf{x}) | y = 1\right]$. $\square$

## A.2 PROOF OF LEMMA 2

*Proof.* Define $\alpha_{*,k} = \arg\max_\alpha f(\bar{\mathbf{v}}_k, \alpha)$, $\mathbf{u} = (\mathbf{v}^\top, \alpha)^\top \in \mathbb{R}^{d+3}$, $\mathbf{u}_{*,k} = (\mathbf{v}_*^\top, \alpha_{*,k})^\top$, $\mathbf{u}_t^k = ((\mathbf{v}_t^k)^\top, \alpha_t^k)^\top$, $\mathbf{g}_t^k = \left(\nabla_\mathbf{v} f(\mathbf{v}_t^k, \alpha_t^k)^\top + \frac{1}{\gamma}\left(\mathbf{v}_t^k - \mathbf{v}_0^k\right)^\top, -\nabla_\alpha f(\mathbf{v}_t^k, \alpha_t^k)^\top\right)^\top$.

$$
\phi_k(\bar{\mathbf{v}}_k) - \min_\mathbf{v} \phi_k(\mathbf{v}) \overset{(a)}{=} \max_\alpha\left[f(\bar{\mathbf{v}}_k, \alpha) + \frac{1}{2\gamma}\|\bar{\mathbf{v}}_k - \bar{\mathbf{v}}_{k-1}\|^2\right] - \min_\mathbf{v} \max_\alpha\left[f(\mathbf{v}, \alpha) + \frac{1}{2\gamma}\|\mathbf{v} - \bar{\mathbf{v}}_{k-1}\|^2\right]
$$

$$
\overset{(b)}{\leq} \left[f(\bar{\mathbf{v}}_k, \alpha_{*,k}) + \frac{1}{2\gamma}\|\bar{\mathbf{v}}_k - \bar{\mathbf{v}}_{k-1}\|^2\right] - \left[f(\mathbf{s}_k, \bar{\alpha}_k) + \frac{1}{2\gamma}\|\mathbf{s}_k - \bar{\mathbf{v}}_{k-1}\|^2\right]
$$

$$
\overset{(c)}{\leq} \frac{\|\bar{\mathbf{v}}_{k-1} - \mathbf{s}_k\|^2}{2\eta_k T_k} + \frac{\|\bar{\alpha}_{k-1} - \alpha_{*,k}\|^2}{2\eta_k T_k} + \eta_k G^2 + \frac{\sum_{t=0}^{T_k-1}(\mathbf{u}_t^k - \mathbf{u}_{*,k})^\top(\mathbf{g}_t^k - \hat{\mathbf{g}}_t^k)}{T_k},
$$

where (a) comes from the definition of $\phi_k$, (b) holds because $\min_\mathbf{v} \max_\alpha\left[f(\mathbf{v}, \alpha) + \frac{1}{2\gamma}\|\mathbf{v} - \bar{\mathbf{v}}_{k-1}\|^2\right] \geq f(\mathbf{s}_k, \bar{\alpha}_k) + \frac{1}{2\gamma}\|\mathbf{s}_k - \bar{\mathbf{v}}_{k-1}\|^2$, (c) comes from the standard analysis of primal-dual stochastic gradient method.

Denote $\mathbb{E}_{k-1}$ by taking the conditional expectation conditioning on all the stochastic events until $\bar{\mathbf{v}}_{k-1}$ is generated. Taking $\mathbb{E}_{k-1}$ on both sides and noting that $\hat{\mathbf{g}}_t^k$ is an unbiased estimator of $\mathbf{g}_t^k$ for $\forall t, k$, we have

$$
\mathbb{E}_{k-1}\left[\phi_k(\bar{\mathbf{v}}_k) - \min_\mathbf{v} \phi_k(\mathbf{v})\right] \leq \frac{\|\bar{\mathbf{v}}_{k-1} - \mathbf{s}_k\|^2}{2\eta_k T_k} + \frac{\mathbb{E}_{k-1}\|\bar{\alpha}_{k-1} - \alpha_{*,k}\|^2}{2\eta_k T_k} + \eta_k G^2 + \mathbf{I},
$$

where

$$
\mathbf{I} = \mathbb{E}_{k-1}\left[\frac{\sum_{t=0}^{T_k-1}(\alpha_t^k - \alpha_{*,k})\left(-\nabla_\alpha f(\mathbf{v}_t^k, \alpha_t^k) - (-\nabla_\alpha F(\mathbf{v}_t^k, \alpha_t^k; \xi_t^k))\right)}{T_k}\right].
$$

Define $\tilde{\alpha}_0^k = \alpha_0^k$ and

$$
\tilde{\alpha}_{t+1}^k = \arg\min_\alpha \left(-\nabla_\alpha f(\mathbf{v}_t^k, \alpha_t^k) - (-\nabla_\alpha F(\mathbf{v}_t^k, \alpha_t^k; \xi_t^k))\right)\alpha + \frac{1}{2\eta_k}(\alpha - \tilde{\alpha}_t^k).
$$

By first-order optimality condition, we have

$$
\left(\tilde{\alpha}_t^k - \alpha_{*,k}\right)\left(-\nabla_\alpha f(\mathbf{v}_t^k, \alpha_t^k) - (-\nabla_\alpha F(\mathbf{v}_t^k, \alpha_t^k; \xi_t^k))\right)
$$
$$
\leq \frac{(\tilde{\alpha}_t^k - \alpha_{*,k})^2 - (\tilde{\alpha}_{t+1}^k - \alpha_{*,k})^2}{2\eta_k} + \frac{\eta_k}{2}\left(-\nabla_\alpha f(\mathbf{v}_t^k, \alpha_t^k) - (-\nabla_\alpha F(\mathbf{v}_t^k, \alpha_t^k; \xi_t^k))\right)^2 \tag{8}
$$

Note that

$$
\mathbf{I} = \mathbb{E}_{k-1}\left[\frac{\sum_{t=0}^{T_k-1}(\alpha_t^k - \tilde{\alpha}_t^k + \tilde{\alpha}_t^k - \alpha_{*,k})\left(-\nabla_\alpha f(\mathbf{v}_t^k, \alpha_t^k) - (-\nabla_\alpha F(\mathbf{v}_t^k, \alpha_t^k; \xi_t^k))\right)}{T_k}\right]
$$

$$
= \mathbb{E}_{k-1}\left[\frac{\sum_{t=0}^{T_k-1}(\tilde{\alpha}_t^k - \alpha_{*,k})\left(-\nabla_\alpha f(\mathbf{v}_t^k, \alpha_t^k) - (-\nabla_\alpha F(\mathbf{v}_t^k, \alpha_t^k; \xi_t^k))\right)}{T_k}\right]
$$

$$
\leq \frac{\mathbb{E}_{k-1}(\tilde{\alpha}_0^k - \alpha_{*,k})^2}{2\eta_k T_k} + \eta_k G^2 = \frac{\mathbb{E}_{k-1}(\bar{\alpha}_{t-1} - \alpha_{*,k})^2}{2\eta_k T_k} + \eta_k G^2
$$

where the first inequality holds due to (8). Hence we have

$$
\mathbb{E}_{k-1}\left[\phi_k(\bar{\mathbf{v}}_k) - \min_\mathbf{v} \phi_k(\mathbf{v})\right] \leq \frac{\|\bar{\mathbf{v}}_{k-1} - \mathbf{s}_k\|^2}{2\eta_k T_k} + \frac{\mathbb{E}_{k-1}\|\bar{\alpha}_{k-1} - \alpha_{*,k}\|^2}{\eta_k T_k} + 2\eta_k G^2.
$$

Define $\mathbf{x}_{j:j+m_{k-1}-1} = (\mathbf{x}_j, \ldots, \mathbf{x}_{j+m_{k-1}-1})$, $y_{j:j+m_{k-1}-1} = (y_j, \ldots, y_{j+m_{k-1}-1})$, and $\tilde{f}(\mathbf{x}_{j:j+m_{k-1}-1}, y_{j:j+m_{k-1}-1}) = \frac{\sum_{i=j}^{j+m_{k-1}-1} h(\bar{\mathbf{w}}_{k-1}; \mathbf{x}_i)\mathbb{I}_{y_i=y}}{\sum_{i=j}^{j+m_{k-1}-1}\mathbb{I}_{y_i=y}} - \mathbb{E}_\mathbf{x}[h(\bar{\mathbf{w}}_{k-1}; \mathbf{x})|y]$. Note that $0 \leq$

$h \leq 1$. Then we know that

$$\mathbb{E}_{\mathbf{x}_{j:j+m_{k-1}-1}}(\tilde{f}^2(\mathbf{x}_{j:j+m_{k-1}-1}, y_{j:j+m_{k-1}-1})|y_{j:j+m_{k-1}-1})$$

$$\leq \frac{\sigma^2}{\sum_{i=j}^{j+m_{k-1}-1} \mathbb{I}_{y_i=y}} \cdot \mathbb{I}_{\left(\sum_{i=j}^{j+m_{k-1}-1} \mathbb{I}_{y_i=y}>0\right)} + 1 \cdot \mathbb{I}_{\left(\sum_{i=j}^{j+m_{k-1}-1} \mathbb{I}_{y_i=y}=0\right)}.$$

Hence

$$\mathbb{E}_{k-1}\left[\tilde{f}^2(\mathbf{x}_j, \ldots, \mathbf{x}_{j+m_{k-1}-1}, y_j, \ldots, y_{j+m_{k-1}-1})\right]$$

$$= \mathbb{E}_{y_{j:j+m_{k-1}-1}}\left[\mathbb{E}_{\mathbf{x}_{j:j+m_{k-1}-1}}(\tilde{f}^2(\mathbf{x}_{j:j+m_{k-1}-1}, y_{j:j+m_{k-1}-1})|y_{j:j+m_{k-1}-1})\right]$$

$$\leq \mathbb{E}_{y_{j:j+m_{k-1}-1}}\left[\frac{\sigma^2}{\sum_{i=j}^{j+m_{k-1}-1} \mathbb{I}_{y_i=y}} \cdot \mathbb{I}_{\left(\sum_{i=j}^{j+m_{k-1}-1} \mathbb{I}_{y_i=y}>0\right)} + 1 \cdot \mathbb{I}_{\left(\sum_{i=j}^{j+m_{k-1}-1} \mathbb{I}_{y_i=y}=0\right)}\right]$$

$$\leq \frac{\sigma^2}{m_{k-1}\mathrm{Pr}(y_i = y)} + (1 - \mathrm{Pr}(y_i = y))^{m_{k-1}}.$$

Hence we have

$$\mathbb{E}_{k-1}\|\bar{\alpha}_{k-1} - \alpha_{*,k-1}\|^2$$

$$= \mathbb{E}_{k-1}\left[\frac{\sum_{i=j}^{j+m_{k-1}-1} h(\bar{\mathbf{w}}_{k-1}; \mathbf{x}_i)\mathbb{I}_{y_i=-1}}{\sum_{i=j}^{j+m_{k-1}-1} \mathbb{I}_{y_i=-1}} - \mathbb{E}_{\mathbf{x}}[h(\bar{\mathbf{w}}_{k-1}; \mathbf{x})|y = -1]\right.$$

$$\left. + \mathbb{E}_{\mathbf{x}}[h(\bar{\mathbf{w}}_{k-1}; \mathbf{x})|y = 1] - \frac{\sum_{i=j}^{j+m_k-1} h(\bar{\mathbf{w}}_{k-1}; \mathbf{x}_i)\mathbb{I}_{y_i=1}}{\sum_{i=j}^{j+m_{k-1}-1} \mathbb{I}_{y_i=1}}\right]^2$$

$$\leq \frac{2\sigma^2}{m_{k-1}\mathrm{Pr}(y_i = -1)} + 2(1 - \mathrm{Pr}(y_i = -1))^{m_{k-1}} + \frac{2\sigma^2}{m_{k-1}\mathrm{Pr}(y_i = 1)} + 2(1 - \mathrm{Pr}(y_i = 1))^{m_{k-1}}$$

$$= \frac{2\sigma^2}{m_{k-1}p(1-p)} + 2p^{m_{k-1}} + 2(1-p)^{m_{k-1}} \leq 2\left(\frac{\sigma^2}{m_{k-1}p(1-p)} + 2(\max(p, 1-p))^{m_{k-1}}\right)$$

$$\overset{(a)}{\leq} 2\left(\frac{\sigma^2}{m_{k-1}p(1-p)} + \frac{C}{m_{k-1}}\right) \leq \frac{2(\sigma^2 + C)}{m_{k-1}p(1-p)}.$$

where $C = \frac{2}{\ln(\frac{1}{\max(p,1-p)})} \max(p, 1-p)^{\frac{1}{\ln(1/\max(p,1-p))}}$, and (a) holds since the function $x\max(p, 1 - p)^x$ achieves its maximum at point $x = 1/\ln(1/\max(p, 1, 1))$.

By the update of $\bar{\alpha}_{k-1}$, $2\tilde{L}$-Lipschitz continuity of $\mathbb{E}[h(\mathbf{w}; \mathbf{x})|y = -1] - \mathbb{E}[h(\mathbf{w}; \mathbf{x})|y = 1]$, and noting that $\alpha_{*,k} = \mathbb{E}[h(\bar{\mathbf{w}}_k; \mathbf{x})|y = -1] - \mathbb{E}[h(\bar{\mathbf{w}}_k; \mathbf{x})|y = 1]$, we have

$$\mathbb{E}_{k-1}\|\bar{\alpha}_{k-1} - \alpha_{*,k}\|^2 = \mathbb{E}_{k-1}\|\bar{\alpha}_{k-1} - \alpha_{*,k-1} + \alpha_{*,k-1} - \alpha_{*,k}\|^2$$

$$\leq \mathbb{E}_{k-1}\left(2\|\bar{\alpha}_{k-1} - \alpha_{*,k-1}\|^2 + 2\|\alpha_{*,k-1} - \alpha_{*,k}\|^2\right) \leq \frac{4(\sigma^2 + C)}{m_{k-1}p(1-p)} + 8\tilde{L}^2\mathbb{E}_{k-1}\|\bar{\mathbf{v}}_{k-1} - \bar{\mathbf{v}}_k\|^2.$$

Taking $m_{k-1} \geq \frac{2(\sigma^2 + C)}{p(1-p)\eta_k^2 G^2 T_k}$, then we have

$$\mathbb{E}_{k-1}\left[\phi_k(\bar{\mathbf{v}}_k) - \min_{\mathbf{v}} \phi_k(\mathbf{v})\right] \leq \frac{\|\bar{\mathbf{v}}_{k-1} - \mathbf{s}_k\|^2 + 16\tilde{L}^2\mathbb{E}_{k-1}\|\bar{\mathbf{v}}_{k-1} - \bar{\mathbf{v}}_k\|^2}{2\eta_k T_k} + 4\eta_k G^2.$$

$\square$

## A.3 PROOF OF THEOREM 2

*Proof.* Define $\phi_k(\mathbf{v}) = \phi(\mathbf{v}) + \frac{1}{2\gamma}\|\mathbf{v} - \bar{\mathbf{v}}_{k-1}\|^2$. We can see that $\phi_k(\mathbf{v})$ is convex and smooth function since $\gamma \leq 1/L$. The smoothness parameter of $\phi_k$ is $\hat{L} = L + \gamma^{-1}$. Define $\mathbf{s}_k = \arg\min_{\mathbf{v}\in\mathbb{R}^{d+2}} \phi_k(\mathbf{v})$. According to Theorem 2.1.5 of (Nesterov, 2013), we have

$$\|\nabla\phi_k(\bar{\mathbf{v}}_k)\|^2 \leq 2\hat{L}(\phi_k(\bar{\mathbf{v}}_k) - \phi_k(\mathbf{s}_k)). \tag{9}$$

Combining (9) with Lemma 2 yields

$$\mathbb{E}_{k-1}\|\nabla\phi_k(\bar{\mathbf{v}}_k)\|^2 \leq 2\hat{L}\left(\frac{\|\bar{\mathbf{v}}_{k-1}-\mathbf{s}_k\|^2 + 16\tilde{L}^2\mathbb{E}_{k-1}\|\bar{\mathbf{v}}_{k-1}-\bar{\mathbf{v}}_k\|^2}{2\eta_k T_k} + 4\eta_k G^2\right). \qquad (10)$$

Note that $\phi_k(\bar{\mathbf{v}})$ is $(\gamma^{-1}-L)$-strongly convex, and $\gamma = \frac{1}{2L}$, we have

$$\phi_k(\bar{\mathbf{v}}_{k-1}) \geq \phi_k(\mathbf{s}_k) + \frac{L}{2}\|\bar{\mathbf{v}}_{k-1}-\mathbf{s}_k\|^2. \qquad (11)$$

Plugging in $\mathbf{s}_k$ into Lemma 2 and combining (11) yield

$$\mathbb{E}_{k-1}[\phi(\bar{\mathbf{v}}_k) + L\|\bar{\mathbf{v}}_k-\bar{\mathbf{v}}_{k-1}\|^2]$$

$$\leq \phi_k(\bar{\mathbf{v}}_{k-1}) - \frac{L}{2}\|\bar{\mathbf{v}}_{k-1}-\mathbf{s}_k\|^2 + \frac{\|\bar{\mathbf{v}}_{k-1}-\mathbf{s}_k\|^2 + 16\tilde{L}^2\mathbb{E}_{k-1}\|\bar{\mathbf{v}}_{k-1}-\bar{\mathbf{v}}_k\|^2}{2\eta_k T_k} + 4\eta_k G^2.$$

By using $\eta_k T_k L \geq \max(2, 16\tilde{L}^2)$, rearranging the terms, and noting that $\phi_k(\bar{\mathbf{v}}_{k-1}) = \phi(\bar{\mathbf{v}}_{k-1})$, we have

$$\frac{\|\bar{\mathbf{v}}_{k-1}-\mathbf{s}_k\|^2 + 16\tilde{L}^2\mathbb{E}_{k-1}\|\bar{\mathbf{v}}_{k-1}-\bar{\mathbf{v}}_k\|^2}{2\eta_k T_k} \leq \phi(\bar{\mathbf{v}}_{k-1}) - \mathbb{E}_{k-1}\left[\phi(\bar{\mathbf{v}}_k)\right] + 4\eta_k G^2. \qquad (12)$$

Combining (12) and (10) yields

$$\mathbb{E}_{k-1}\|\nabla\phi_k(\bar{\mathbf{v}}_k)\|^2 \leq 6L\left(\phi(\bar{\mathbf{v}}_{k-1}) - \mathbb{E}_{k-1}\left[\phi(\bar{\mathbf{v}}_k)\right] + 8\eta_k G^2\right). \qquad (13)$$

Taking expectation on both sides over all randomness until $\bar{\mathbf{v}}_{k-1}$ is generated and by the tower property, we have

$$\mathbb{E}\|\nabla\phi_k(\bar{\mathbf{v}}_k)\|^2 \leq 6L\left(\mathbb{E}\left[\phi(\bar{\mathbf{v}}_{k-1}) - \phi(\mathbf{v}_*)\right] - \mathbb{E}\left[\phi(\bar{\mathbf{v}}_k) - \phi(\mathbf{v}_*)\right] + 8\eta_k G^2\right). \qquad (14)$$

Note that $\phi(\mathbf{v})$ is $L$-smooth and hence is $L$-weakly convex, so we have

$$\phi(\bar{\mathbf{v}}_{k-1}) \geq \phi(\bar{\mathbf{v}}_k) + \langle\nabla\phi(\bar{\mathbf{v}}_k), \bar{\mathbf{v}}_{k-1}-\bar{\mathbf{v}}_k\rangle - \frac{L}{2}\|\bar{\mathbf{v}}_{k-1}-\bar{\mathbf{v}}_k\|^2$$

$$= \phi(\bar{\mathbf{v}}_k) + \langle\nabla\phi(\bar{\mathbf{v}}_k) + 2L(\bar{\mathbf{v}}_k-\bar{\mathbf{v}}_{k-1}), \bar{\mathbf{v}}_{k-1}-\bar{\mathbf{v}}_k\rangle + \frac{3}{2}L\|\bar{\mathbf{v}}_{k-1}-\bar{\mathbf{v}}_k\|^2$$

$$\overset{(a)}{=} \phi(\bar{\mathbf{v}}_k) + \langle\nabla\phi_k(\bar{\mathbf{v}}_k), \bar{\mathbf{v}}_{k-1}-\bar{\mathbf{v}}_k\rangle + \frac{3}{2}L\|\bar{\mathbf{v}}_{k-1}-\bar{\mathbf{v}}_k\|^2 \qquad (15)$$

$$\overset{(b)}{=} \phi(\bar{\mathbf{v}}_k) - \frac{1}{2L}\langle\nabla\phi_k(\bar{\mathbf{v}}_k), \nabla\phi_k(\bar{\mathbf{v}}_k) - \nabla\phi(\bar{\mathbf{v}}_k)\rangle + \frac{3}{8L}\|\nabla\phi_k(\bar{\mathbf{v}}_k) - \nabla\phi(\bar{\mathbf{v}}_k)\|^2$$

$$= \phi(\bar{\mathbf{v}}_k) - \frac{1}{8L}\|\nabla\phi_k(\bar{\mathbf{v}}_k)\|^2 - \frac{1}{4L}\langle\nabla\phi_k(\bar{\mathbf{v}}_k), \nabla\phi(\bar{\mathbf{v}}_k)\rangle + \frac{3}{8L}\|\nabla\phi(\bar{\mathbf{v}}_k)\|^2,$$

where (a) and (b) hold by the definition of $\phi_k$.

Rearranging the terms in (15) yields

$$\phi(\bar{\mathbf{v}}_k) - \phi(\bar{\mathbf{v}}_{k-1}) \leq \frac{1}{8L}\|\nabla\phi_k(\bar{\mathbf{v}}_k)\|^2 + \frac{1}{4L}\langle\nabla\phi_k(\bar{\mathbf{v}}_k), \nabla\phi(\bar{\mathbf{v}}_k)\rangle - \frac{3}{8L}\|\nabla\phi(\bar{\mathbf{v}}_k)\|^2$$

$$\overset{(a)}{\leq} \frac{1}{8L}\|\nabla\phi_k(\bar{\mathbf{v}}_k)\|^2 + \frac{1}{8L}\left(\|\nabla\phi_k(\bar{\mathbf{v}}_k)\|^2 + \|\nabla\phi(\bar{\mathbf{v}}_k)\|^2\right) - \frac{3}{8L}\|\nabla\phi(\bar{\mathbf{v}}_k)\|^2$$

$$= \frac{1}{4L}\|\nabla\phi_k(\bar{\mathbf{v}}_k)\|^2 - \frac{1}{4L}\|\nabla\phi(\bar{\mathbf{v}}_k)\|^2$$

$$\overset{(b)}{\leq} \frac{1}{4L}\|\nabla\phi_k(\bar{\mathbf{v}}_k)\|^2 - \frac{\mu}{2L}\left(\phi(\bar{\mathbf{v}}_k) - \phi(\mathbf{v}_*)\right),$$

$$\qquad (16)$$

where (a) holds by using $\langle\mathbf{a}, \mathbf{b}\rangle \leq \frac{1}{2}(\|\mathbf{a}\|^2 + \|\mathbf{b}\|^2)$, and (b) holds by the PL property of $\phi$.

Define $\Delta_k = \phi(\bar{\mathbf{v}}_k) - \phi(\mathbf{v}_*)$. Combining (14) and (16), we can see that

$$\mathbb{E}[\Delta_k - \Delta_{k-1}] \leq \frac{3}{2}\left(\mathbb{E}[\Delta_{k-1} - \Delta_k] + 8\eta_k G^2\right) - \frac{\mu}{2L}\mathbb{E}[\Delta_k],$$

which implies that

$$\left(\frac{5}{2} + \frac{\mu}{2L}\right)\mathbb{E}[\Delta_k] \leq \frac{5}{2}\mathbb{E}[\Delta_{k-1}] + 12\eta_k G^2.$$

As a result, we have

$$\mathbb{E}[\Delta_k] \le \frac{5}{5+\mu/L}\mathbb{E}[\Delta_{k-1}] + \frac{24\eta_k G^2}{5+\mu/L} = \left(1 - \frac{\mu/L}{5+\mu/L}\right)\left(\mathbb{E}[\Delta_{k-1}] + \frac{24}{5}\eta_k G^2\right)$$

$$\le \left(1 - \frac{\mu/L}{5+\mu/L}\right)^k \mathbb{E}[\Delta_0] + \frac{24}{5}G^2 \sum_{j=1}^{k} \eta_j \left(1 - \frac{\mu/L}{5+\mu/L}\right)^{k+1-j}.$$

By setting $\eta_k = \eta_0 \exp\left(-(k-1)\frac{\mu/L}{5+\mu/L}\right)$, we have

$$\mathbb{E}[\Delta_k] \le \left(1 - \frac{\mu/L}{5+\mu/L}\right)^k \mathbb{E}[\Delta_0] + \frac{24}{5}G^2\eta_0 \sum_{j=1}^{k} \exp\left(-k\frac{\mu/L}{5+\mu/L}\right)$$

$$\le \exp\left(-k\frac{\mu/L}{5+\mu/L}\right)\Delta_0 + \frac{24}{5}G^2\eta_0 k \exp\left(-k\frac{\mu/L}{5+\mu/L}\right).$$

To achieve $\mathbb{E}[\Delta_k] \le \epsilon$, it suffices to let $K$ satisfy $\exp\left(-K\frac{\mu/L}{5+\mu/L}\right) \le \min\left(\frac{\epsilon}{2\Delta_0}, \frac{5\epsilon}{48KG^2\eta_0}\right)$, i.e.
$K \ge \left(\frac{5L}{\mu}+1\right)\max\left(\log\frac{2\Delta_0}{\epsilon}, \log K + \log\frac{48G^2\eta_0}{5\epsilon}\right)$.

Since $\eta_k T_k L \ge \max(2, 16\tilde{L}^2)$, by the setting of $\eta_k$, we set $T_k = \frac{\max(2,16\tilde{L}^2)}{L\eta_0}\exp\left((k-1)\frac{\mu/L}{5+\mu/L}\right)$.
Then the total iteration complexity is

$$\sum_{k=1}^{K} T_k \le \frac{\max(2, 16\tilde{L}^2)}{L\eta_0} \cdot \frac{\exp\left(K\frac{\mu/L}{5+\mu/L}\right) - 1}{\exp\left(\frac{\mu/L}{5+\mu/L}\right) - 1} = \widetilde{O}\left(\frac{KG^2}{\mu\epsilon}\right) = \widetilde{O}\left(\frac{LG^2}{\mu^2\epsilon}\right).$$

The required number of samples is

$$\sum_{k=1}^{K} m_k = \frac{2(\sigma^2 + C)L}{p(1-p)G^2\eta_0 \max(2, 16\tilde{L}^2)} \cdot \frac{\exp\left(K\frac{\mu/L}{5+\mu/L}\right) - 1}{\exp\left(\frac{\mu/L}{5+\mu/L}\right) - 1} = \widetilde{O}\left(\frac{L^3\sigma^2}{\mu^2\epsilon}\right).$$

$\square$

### A.4 PROOF OF LEMMA 3

*Proof.* Define $\alpha_{*,k} = \arg\max_\alpha f(\bar{\mathbf{v}}_k, \alpha)$, $\mathbf{u} = (\mathbf{v}^\top, \alpha)^\top \in \mathbb{R}^{d+3}$, $\mathbf{u}_{*,k} = (\mathbf{v}_*^\top, \alpha_{*,k})^\top$, $\mathbf{u}_t^k = ((\mathbf{v}_t^k)^\top, \alpha_t^k)^\top$.

$$\phi_k(\bar{\mathbf{v}}_k) - \min_{\mathbf{v}}\phi_k(\mathbf{v}) \overset{(a)}{=} \max_\alpha\left[f(\bar{\mathbf{v}}_k, \alpha) + \frac{1}{2\gamma}\|\bar{\mathbf{v}}_k - \bar{\mathbf{v}}_{k-1}\|^2\right] - \min_{\mathbf{v}}\max_\alpha\left[f(\mathbf{v}, \alpha) + \frac{1}{2\gamma}\|\mathbf{v} - \bar{\mathbf{v}}_{k-1}\|^2\right]$$

$$\overset{(b)}{\le} \left[f(\bar{\mathbf{v}}_k, \alpha_{*,k}) + \frac{1}{2\gamma}\|\bar{\mathbf{v}}_k - \bar{\mathbf{v}}_{k-1}\|^2\right] - \left[f(\mathbf{s}_k, \bar{\alpha}_k) + \frac{1}{2\gamma}\|\mathbf{s}_k - \bar{\mathbf{v}}_{k-1}\|^2\right]$$

$$\overset{(c)}{\le} \frac{1}{T_k}\sum_{t=1}^{T_k}\left[f(\mathbf{v}_t^k, \alpha_{*,k}) + \frac{1}{2\gamma}\|\mathbf{v}_t^k - \bar{\mathbf{v}}_{k-1}\|^2 - \left(f(\mathbf{s}_k, \alpha_t^k) + \frac{1}{2\gamma}\|\mathbf{s}_k - \bar{\mathbf{v}}_{k-1}\|^2\right)\right]$$

$$= \frac{1}{T_k}\sum_{t=1}^{T_k}\left[\left(f(\mathbf{v}_t^k, \alpha_{*,k}) + \frac{1}{2\gamma}\|\mathbf{v}_t^k - \bar{\mathbf{v}}_{k-1}\|^2\right) - \left(f(\mathbf{v}_t^k, \alpha_t^k) + \frac{1}{2\gamma}\|\mathbf{v}_t^k - \bar{\mathbf{v}}_{k-1}\|^2\right)\right.$$

$$\left. + \left(f(\mathbf{v}_t^k, \alpha_t^k) + \frac{1}{2\gamma}\|\mathbf{v}_t^k - \bar{\mathbf{v}}_{k-1}\|^2\right) - \left(f(\mathbf{s}_k, \alpha_t^k) + \frac{1}{2\gamma}\|\mathbf{s}_k - \bar{\mathbf{v}}_{k-1}\|^2\right)\right]$$

$$\le \frac{1}{T_k}\sum_{t=1}^{T_k}\left\langle \nabla_{\mathbf{v}}\left(f(\mathbf{v}_t^k, \alpha_t^k) + \frac{1}{2\gamma}\|\mathbf{v}_t^k - \mathbf{v}_t^0\|^2\right), \mathbf{v}_t^k - \mathbf{s}_k\right\rangle + \left\langle -\nabla_\alpha\left(f(\mathbf{v}_t^k, \alpha_t^k) + \frac{1}{2\gamma}\|\mathbf{v}_t^k - \mathbf{v}_t^0\|^2\right), \alpha_t^k - \alpha_{*,k}\right\rangle$$

$$= \frac{\sum_{t=1}^{T_k}\left\langle \mathbf{u}_t^k - \mathbf{u}_{*,k}, \hat{\mathbf{g}}_t^k\right\rangle}{T_k} + \frac{\sum_{t=1}^{T_k}\left\langle \mathbf{u}_t^k - \mathbf{u}_{*,k}, \mathbf{g}_t^k - \hat{\mathbf{g}}_t^k\right\rangle}{T_k}$$

$$= \mathbf{I} + \mathbf{II}$$

(17)

where (a) comes from the definition of $\phi_k$, (b) holds because $\min_{\mathbf{v}} \max_{\alpha} \left[ f(\mathbf{v}, \alpha) + \frac{1}{2\gamma} \|\mathbf{v} - \bar{\mathbf{v}}_{k-1}\|^2 \right] \geq f(\mathbf{s}_k, \bar{\alpha}_k) + \frac{1}{2\gamma} \|\mathbf{s}_k - \bar{\mathbf{v}}_{k-1}\|^2$, (c) holds by Jensen's inequality.

Now we bound $\mathbf{I}$ and $\mathbf{II}$ separately. Define $\|\mathbf{u}\|_H = \sqrt{\mathbf{u}^\top H \mathbf{u}}$, $\psi_0^k(\mathbf{u}) = 0$, $\psi_{T_k}^{k,*}$ to be the conjugate of $\frac{1}{\eta_k} \psi_{T_k}^k$, which is $\psi_{T_k}^{k,*}(\mathbf{g}) = \sup_{\mathbf{u}} \left\{ \langle \mathbf{g}, \mathbf{u} \rangle - \frac{1}{\eta_k} \psi_{T_k}^k(\mathbf{u}) \right\}$. Note that

$$
\begin{aligned}
T_k \cdot \mathbf{I} &= \sum_{t=1}^{T_k} \langle \hat{\mathbf{g}}_t^k, \mathbf{u}_t^k \rangle - \sum_{t=1}^{T_k} \langle \hat{\mathbf{g}}_t^k, \mathbf{u}_{*,k} \rangle - \frac{1}{\eta_k} \psi_{T_k}^k(\mathbf{u}_{*,k}) + \frac{1}{\eta_k} \psi_{T_k}^k(\mathbf{u}_{*,k}) \\
&\leq \frac{1}{\eta_k} \psi_{T_k}^k(\mathbf{u}_{*,k}) + \sum_{t=1}^{T_k} \langle \hat{\mathbf{g}}_t^k, \mathbf{u}_t^k \rangle + \sup_{\mathbf{u}} \left\{ \left\langle -\sum_{t=1}^{T_k} \hat{\mathbf{g}}_t^k, \mathbf{u} \right\rangle - \frac{1}{\eta_k} \psi_{T_k}^k(\mathbf{u}_{*,k}) \right\} \quad (18) \\
&= \frac{1}{\eta_k} \psi_{T_k}^k(\mathbf{u}_{*,k}) + \sum_{t=1}^{T_k} \langle \hat{\mathbf{g}}_t^k, \mathbf{u}_t^k \rangle + \psi_{T_k}^{k,*} \left( -\sum_{t=1}^{T_k} \hat{\mathbf{g}}_t^k \right),
\end{aligned}
$$

where the last equality holds by the definition of $\psi_{T_k}^{k,*}$.

In addition, note that

$$
\begin{aligned}
\psi_{T_k}^{k,*} \left( -\sum_{t=1}^{T_k} \hat{\mathbf{g}}_t^k \right) &\overset{(a)}{=} \left\langle -\sum_{t=1}^{T_k} \hat{\mathbf{g}}_t^k, \mathbf{u}_{T_k+1}^k \right\rangle - \frac{1}{\eta_k} \psi_{T_k}^k(\mathbf{u}_{T_k+1}^k) \overset{(b)}{\leq} \left\langle -\sum_{t=1}^{T_k} \hat{\mathbf{g}}_t^k, \mathbf{u}_{T_k+1} \right\rangle - \frac{1}{\eta_k} \psi_{T_k-1}^k(\mathbf{u}_{T_k+1}^k) \\
&\leq \sup_{\mathbf{u}} \left\{ \left\langle -\sum_{t=1}^{T_k} \hat{\mathbf{g}}_k^t, \mathbf{u} \right\rangle - \frac{1}{\eta_k} \psi_{T_k-1}^k(\mathbf{u}) \right\} = \psi_{T_k-1}^{k,*} \left( -\sum_{t=1}^{T_k} \hat{\mathbf{g}}_t^k \right) \\
&\overset{(c)}{\leq} \psi_{T_k-1}^{k,*} \left( -\sum_{t=1}^{T_k-1} \hat{\mathbf{g}}_t^k \right) + \left\langle -\mathbf{g}_{T_k}^k, \nabla \psi_{T_k-1}^{k,*} \left( -\sum_{t=1}^{T_k-1} \hat{\mathbf{g}}_t^k \right) \right\rangle + \frac{\eta_k}{2} \|\hat{\mathbf{g}}_{T_k}\|_{\psi_{T_k-1}^{k,*}}^2,
\end{aligned}
$$
(19)

where (a) holds due to the update of the Algorithm 3, (b) holds since $\psi_{t+1}^k(\mathbf{u}) \geq \psi_t^k(\mathbf{u})$, (c) holds by the $\eta_k$-smoothness of $\psi_t^{k,*}$ with respect to $\|\cdot\|_{\psi_t^{k,*}} = \|\cdot\|_{(H_t^k)^{-1}}$.

By (19) and noting that $\nabla \psi_{T_k-1}^{k,*} \left( -\sum_{t=1}^{T_k-1} \hat{\mathbf{g}}_t^k \right) = \mathbf{u}_{T_k}^k$, we have

$$
\sum_{t=1}^{T_k} \langle \hat{\mathbf{g}}_t^k, \mathbf{u}_t^k \rangle + \psi_{T_k}^{k,*} \left( -\sum_{t=1}^{T_k} \hat{\mathbf{g}}_t^k \right) \leq \sum_{t=1}^{T_k-1} \langle \hat{\mathbf{g}}_t^k, \mathbf{u}_t^k \rangle + \psi_{T_k-1}^{k,*} \left( -\sum_{t=1}^{T_k-1} \hat{\mathbf{g}}_t^k \right) + \frac{\eta_k}{2} \|\hat{\mathbf{g}}_{T_k}\|_{\psi_{T_k-1}^{k,*}}^2
$$
(20)

Using (20) recursively and noting that $\psi_0^k(\mathbf{u}) = 0$, we know that

$$
\sum_{t=1}^{T_k} \langle \hat{\mathbf{g}}_t^k, \mathbf{u}_t^k \rangle + \psi_{T_k}^{k,*} \left( -\sum_{t=1}^{T_k} \hat{\mathbf{g}}_t^k \right) \leq \frac{\eta_k}{2} \sum_{t=1}^{T_k} \|\hat{\mathbf{g}}_t^k\|_{\psi_{t-1}^{k,*}}^2
$$
(21)

Combining (18) and (21), we have

$$
\mathbf{I} \leq \frac{1}{\eta_k T_k} \psi_{T_k}^k(\mathbf{u}_{*,k}) + \frac{\eta_k}{2T_k} \sum_{t=1}^{T_k} \|\hat{\mathbf{g}}_t^k\|_{\psi_{t-1}^{k,*}}^2
$$
(22)

By Lemma 4 of (Duchi et al., 2011) and setting $\delta \geq \max_t \|\hat{\mathbf{g}}_t^k\|_\infty$, we know that $\sum_{t=1}^{T_k} \|\hat{\mathbf{g}}_t^k\|_{\psi_{t-1}^{k,*}}^2 \leq 2\sum_{i=1}^{d+3} \|\hat{\mathbf{g}}_{1:T_k}^k\|_2$, and hence

$$
\begin{aligned}
\mathbf{I} &\leq \frac{1}{\eta_k T_k} \psi_{T_k}^k(\mathbf{u}_{*,k}) + \frac{\eta_k}{T_k} \sum_{i=1}^{d+3} \|\hat{\mathbf{g}}_{1:T_k}^k\|_2 \\
&= \frac{\delta \|\mathbf{u}_1^k - \mathbf{u}_{*,k}\|_2^2}{2\eta_k T_k} + \frac{\langle \mathbf{u}_1^k - \mathbf{u}_{*,k}, \operatorname{diag}(s_{T_k}^k)(\mathbf{u}_1^k - \mathbf{u}_{*,k})\rangle}{2\eta_k T_k} + \frac{\eta_k}{T_k} \sum_{i=1}^d \|\hat{\mathbf{g}}_{1:T_k}^k\|_2 \quad (23) \\
&\leq \frac{\delta + \max_i \|\hat{\mathbf{g}}_{1:T_k,i}^k\|_2}{2\eta_k T_k} \|\mathbf{u}_1^k - \mathbf{u}_{*,k}\|_2^2 + \frac{\eta_k}{T_k} \sum_{i=1}^{d+3} \|\hat{\mathbf{g}}_{1:T_k}^k\|_2
\end{aligned}
$$

Denote $\mathbb{E}_{k-1}$ by taking the conditional expectation conditioning on filtration $\mathcal{F}_{k-1}$, where $\mathcal{F}_{k-1}$ is the $\sigma$-algebra generated by all random variables until $\bar{\mathbf{v}}_{k-1}$ is generated. Taking $\mathbb{E}_{k-1}$ on both sides of (17), and employing (23) yields

$$
\begin{aligned}
&\mathbb{E}_{k-1}\left[\phi_k(\bar{\mathbf{v}}_k) - \min_{\mathbf{v}} \phi_k(\mathbf{v})\right] \\
&\leq \mathbb{E}_{k-1}\left[\frac{\delta + \max_i \|\hat{\mathbf{g}}_{1:T_k,i}^k\|_2}{2\eta_k T_k}\left(\|\bar{\mathbf{v}}_{k-1} - \mathbf{s}_k\|_2^2 + \|\bar{\alpha}_{k-1} - \alpha_{*,k}\|_2^2\right) + \frac{\eta_k}{T_k} \sum_{i=1}^{d+3} \|\hat{\mathbf{g}}_{1:T_k}^k\|_2\right] + \mathbb{E}_{k-1}(\mathbf{II}) \\
&= \left(\|\bar{\mathbf{v}}_{k-1} - \mathbf{s}_k\|_2^2\right)\mathbb{E}_{k-1}\left(\frac{\delta + \max_i \|\hat{\mathbf{g}}_{1:T_k,i}^k\|_2}{2\eta_k T_k}\right) + \mathbb{E}_{k-1}\left(\frac{\delta + \max_i \|\hat{\mathbf{g}}_{1:T_k,i}^k\|_2}{2\eta_k T_k} \|\bar{\alpha}_{k-1} - \alpha_{*,k}\|_2^2\right) \\
&\quad + \mathbb{E}_{k-1}\left(\frac{\eta_k}{T_k} \sum_{i=1}^{d+3} \|\hat{\mathbf{g}}_{1:T_k}^k\|_2\right) + \mathbb{E}_{k-1}(\mathbf{II})
\end{aligned}
$$

$$(24)$$

where the equality holds since $\bar{\mathbf{v}}_{k-1} - \mathbf{s}_k$ is measurable with respect to $\mathcal{F}_{k-1}$.

Note that

$$
\begin{aligned}
&\mathbb{E}_{k-1}\left(\frac{\delta + \max_i \|\hat{\mathbf{g}}_{1:T_k,i}^k\|_2}{2\eta_k T_k} \|\bar{\alpha}_{k-1} - \alpha_{*,k}\|_2^2\right) = \mathbb{E}_{k-1}\left(\frac{\delta + \max_i \|\hat{\mathbf{g}}_{1:T_k,i}^k\|_2}{2\eta_k T_k} \|\bar{\alpha}_{k-1} - \alpha_{*,k-1} + \alpha_{*,k-1} - \alpha_{*,k}\|^2\right) \\
&\leq \mathbb{E}_{k-1}\left(\frac{\delta + \max_i \|\hat{\mathbf{g}}_{1:T_k,i}^k\|_2}{2\eta_k T_k} \left(2\|\bar{\alpha}_{k-1} - \alpha_{*,k-1}\|^2 + 2\|\alpha_{*,k-1} - \alpha_{*,k}\|^2\right)\right) \\
&\overset{(a)}{=} \mathbb{E}_{k-1}\left(\frac{\delta + \max_i \|\hat{\mathbf{g}}_{1:T_k,i}^k\|_2}{2\eta_k T_k}\right)\mathbb{E}_{k-1}\left(2\|\bar{\alpha}_{k-1} - \alpha_{*,k-1}\|^2\right) + \mathbb{E}_{k-1}\left(\frac{\delta + \max_i \|\hat{\mathbf{g}}_{1:T_k,i}^k\|_2}{2\eta_k T_k} \cdot 2\|\alpha_{*,k-1} - \alpha_{*,k}\|^2\right) \\
&\overset{(b)}{\leq} \mathbb{E}_{k-1}\left(\frac{\delta + \max_i \|\hat{\mathbf{g}}_{1:T_k,i}^k\|_2}{2\eta_k T_k}\right) \cdot \frac{4(\sigma^2 + C)}{m_{k-1}p(1-p)} + \mathbb{E}_{k-1}\left(\frac{\delta + \max_i \|\hat{\mathbf{g}}_{1:T_k,i}^k\|_2}{2\eta_k T_k} \cdot 2\|\alpha_{*,k-1} - \alpha_{*,k}\|^2\right) \\
&\overset{(c)}{\leq} \mathbb{E}_{k-1}\left(\frac{\delta + \max_i \|\hat{\mathbf{g}}_{1:T_k,i}^k\|_2}{2\eta_k T_k}\right) \cdot \frac{4(\sigma^2 + C)}{m_{k-1}p(1-p)} + \mathbb{E}_{k-1}\left(\frac{\delta + \max_i \|\hat{\mathbf{g}}_{1:T_k,i}^k\|_2}{2\eta_k T_k} \cdot 8\tilde{L}^2\|\bar{\mathbf{v}}_{k-1} - \bar{\mathbf{v}}_k\|^2\right)
\end{aligned}
$$

where $(a)$ holds because $\bar{\alpha}_{k-1} - \alpha_{*,k-1}$ and $\frac{\delta + \max_i \|\hat{\mathbf{g}}_{1:T_k,i}^k\|_2}{2\eta_k T_k}$ are independent conditioning on $\mathcal{F}_{k-1}$, (b) holds because of the update of $\bar{\alpha}_{k-1}$ and $\alpha_{*,k} = \mathbb{E}\left[h(\bar{\mathbf{w}}_k; \mathbf{x})|y = -1\right] - \mathbb{E}\left[h(\bar{\mathbf{w}}_k; \mathbf{x})|y = 1\right]$, (c) holds due to the $2\tilde{L}$-Lipschitz continuity of $\mathbb{E}\left[h(\mathbf{w}; \mathbf{x})|y = -1\right] - \mathbb{E}\left[h(\mathbf{w}; \mathbf{x})|y = 1\right]$.

Taking $m_{k-1} \geq \frac{2(\sigma^2 + C)}{p(1-p)(d+3)\eta_k^2}$, then we have

$$\mathbb{E}_{k-1}\left[\phi_k(\bar{\mathbf{v}}_k) - \min_{\mathbf{v}} \phi_k(\mathbf{v})\right]$$

$$\leq \mathbb{E}_{k-1}\left(\frac{\delta + \max_i \|\hat{\mathbf{g}}_{1:T_k,i}^k\|_2}{2\eta_k T_k}\right) \|\bar{\mathbf{v}}_{k-1} - \mathbf{s}_k\|^2 + \mathbb{E}_{k-1}\left(\frac{\eta_k}{T_k}\sum_{i=1}^{d+3} \|\hat{\mathbf{g}}_{1:T_k,i}^k\|_2\right) + \mathbb{E}_{k-1}(\mathbf{II})$$

$$+ \mathbb{E}_{k-1}\left(\frac{\delta + \max_i \|\hat{\mathbf{g}}_{1:T_k,i}^k\|_2}{2T_k}\right) \cdot 2\eta_k(d+3) + \mathbb{E}_{k-1}\left(\frac{\delta + \max_i \|\hat{\mathbf{g}}_{1:T_k,i}^k\|_2}{2\eta_k T_k} \cdot 8\tilde{L}^2 \|\bar{\mathbf{v}}_{k-1} - \bar{\mathbf{v}}_k\|^2\right)$$

$$= \mathbb{E}_{k-1}\left(\frac{\delta + \max_i \|\hat{\mathbf{g}}_{1:T_k,i}^k\|_2}{2\eta_k T_k}\right) \|\bar{\mathbf{v}}_{k-1} - \mathbf{s}_k\|^2 + \mathbb{E}_{k-1}\left[\frac{\eta_k}{T_k}\left(\sum_{i=1}^{d+3} \|\hat{\mathbf{g}}_{1:T_k,i}^k\|_2 + (d+3)\left(\delta + \max_i \|\hat{\mathbf{g}}_{1:T_k,i}^k\|_2\right)\right)\right]$$

$$+ \mathbb{E}_{k-1}(\mathbf{II}) + \mathbb{E}_{k-1}\left(\frac{\delta + \max_i \|\hat{\mathbf{g}}_{1:T_k,i}^k\|_2}{2\eta_k T_k} \cdot 8\tilde{L}^2 \|\bar{\mathbf{v}}_{k-1} - \bar{\mathbf{v}}_k\|^2\right)$$

Define $\widetilde{\alpha}_0^k = \alpha_0^k$ and

$$\widetilde{\alpha}_{t+1}^k = \arg\min_\alpha \left[\frac{\eta_k}{t}\sum_{\tau=1}^{t}\left(-\nabla_\alpha f(\mathbf{v}_t^k, \alpha_t^k) - (-\nabla_\alpha F(\mathbf{v}_t^k, \alpha_t^k; \xi_t^k))\right)\alpha + \frac{1}{t}\psi_t^k(\alpha)\right],$$

where $\psi_t^k(\alpha) = \psi_t^k(\mathbf{u})$ in which $\mathbf{u} = [0, \ldots, 0, \alpha]$ and $\mathbf{u}_0^k = [0, \ldots, 0, \alpha_0^k]$. By setting

$$T_k = \inf\left\{\tau : \tau \geq M_k \max\left(\frac{(\delta + \max_i \|\hat{\mathbf{g}}_{1:\tau,i}^k\|_2)\max(1, 8\tilde{L}^2)}{c}, 2c\left(\sum_{i=1}^{d+3}\|\hat{\mathbf{g}}_{1:\tau,i}^k\|_2 + (d+3)\left(\delta + \max_i \|\hat{\mathbf{g}}_{1:\tau,i}^k\|_2\right)\right)\right)\right\},$$

then $T_k$ is a stopping time which is bounded almost surely. By stopping time argument, we have

$$\mathbb{E}_{k-1}\left[\frac{\sum_{t=1}^{T_k}(\mathbf{v}_t^k - \mathbf{v}_*)^\top\left(\nabla_{\mathbf{v}} f(\mathbf{v}_t^k, \alpha_t^k) - \nabla_{\mathbf{v}} F(\mathbf{v}_t^k, \alpha_t^k; \xi_t^k)\right)}{T_k}\right] = 0$$

$$\mathbb{E}_{k-1}\left[\frac{\sum_{t=1}^{T_k}(\alpha_t^k - \widetilde{\alpha}_t^k)^\top\left(-\nabla_\alpha f(\mathbf{v}_t^k, \alpha_t^k) - (-\nabla_\alpha F(\mathbf{v}_t^k, \alpha_t^k; \xi_t^k))\right)}{T_k}\right] = 0$$

Hence we know that

$$\mathbb{E}_{k-1}(\mathbf{II}) = \mathbb{E}_{k-1}\left[\frac{\sum_{t=1}^{T_k}(\widetilde{\alpha}_t^k - \alpha_{*,k})\left(-\nabla_\alpha f(\mathbf{v}_t^k, \alpha_t^k) - (-\nabla_\alpha F(\mathbf{v}_t^k, \alpha_t^k; \xi_t^k))\right)}{T_k}\right].$$

Note that the variance of stochastic gradient is smaller than its second moment, we can follow the similar analysis of bounding $\mathbf{I}$ to show that

$$\mathbb{E}_{k-1}(\mathbf{II}) \leq \mathbb{E}_{k-1}\left[\frac{\delta + \max_i \|\hat{\mathbf{g}}_{1:T_k,i}^k\|_2}{2\eta_k T_k}\|\mathbf{u}_1^k - \mathbf{u}_{*,k}\|_2^2 + \frac{\eta_k}{T_k}\sum_{i=1}^{d+3}\|\hat{\mathbf{g}}_{1:T_k}^k\|_2\right].$$

Following the same analysis of bounding the RHS of (23), we know that

$$\mathbb{E}_{k-1}\left[\phi_k(\bar{\mathbf{v}}_k) - \min_{\mathbf{v}} \phi_k(\mathbf{v})\right] \leq \frac{c\left(\|\bar{\mathbf{v}}_{k-1} - \mathbf{s}_k\|_2^2 + \mathbb{E}_{k-1}\|\bar{\mathbf{v}}_{k-1} - \bar{\mathbf{v}}_k\|_2^2\right)}{\eta_k M_k} + \frac{\eta_k}{cM_k}.$$

$\square$

## A.5 PROOF OF THEOREM 3

*Proof.* Define $\phi_k(\mathbf{v}) = \phi(\mathbf{v}) + \frac{1}{2\gamma}\|\mathbf{v} - \bar{\mathbf{v}}_{k-1}\|^2$. We can see that $\phi_k(\mathbf{v})$ is convex and smooth function since $\gamma \leq 1/L$. The smoothness parameter of $\phi_k$ is $\hat{L} = L + \gamma^{-1}$. Define $\mathbf{s}_k = \arg\min_{\mathbf{v} \in \mathbb{R}^{d+2}} \phi_k(\mathbf{v})$. According to Theorem 2.1.5 of (Nesterov, 2013), we have

$$\|\nabla \phi_k(\bar{\mathbf{v}}_k)\|^2 \leq 2\hat{L}(\phi_k(\bar{\mathbf{v}}_k) - \phi_k(\mathbf{s}_k)). \tag{25}$$

Combining (25) with Lemma 3 yields

$$\mathbb{E}_{k-1}\|\nabla\phi_k(\bar{\mathbf{v}}_k)\|^2 \leq 2\hat{L}\left(\frac{c\left(\|\bar{\mathbf{v}}_{k-1} - \mathbf{s}_k\|_2^2 + \mathbb{E}_{k-1}\|\bar{\mathbf{v}}_{k-1} - \bar{\mathbf{v}}_k\|_2^2\right)}{\eta_k M_k} + \frac{\eta_k}{cM_k}\right). \qquad (26)$$

Note that $\phi_k(\bar{\mathbf{v}})$ is $(\gamma^{-1} - L)$-strongly convex, and $\gamma = \frac{1}{2L}$, we have

$$\phi_k(\bar{\mathbf{v}}_{k-1}) \geq \phi_k(\mathbf{s}_k) + \frac{L}{2}\|\bar{\mathbf{v}}_{k-1} - \mathbf{s}_k\|^2. \qquad (27)$$

Plugging in $\mathbf{s}_k$ into Lemma 3 and combining (27) yield

$$\mathbb{E}_{k-1}[\phi(\bar{\mathbf{v}}_k) + L\|\bar{\mathbf{v}}_k - \bar{\mathbf{v}}_{k-1}\|^2] \leq \phi_k(\bar{\mathbf{v}}_{k-1}) - \frac{L}{2}\|\bar{\mathbf{v}}_{k-1} - \mathbf{s}_k\|^2 + \frac{c\left(\|\bar{\mathbf{v}}_{k-1} - \mathbf{s}_k\|_2^2 + \mathbb{E}_{k-1}\|\bar{\mathbf{v}}_{k-1} - \bar{\mathbf{v}}_k\|_2^2\right)}{\eta_k M_k} + \frac{\eta_k}{cM_k}$$

By taking $\eta_k M_k L \geq 4c$, rearranging the terms, and noting that $\phi_k(\bar{\mathbf{v}}_{k-1}) = \phi(\bar{\mathbf{v}}_{k-1})$, we have

$$\frac{c\left(\|\bar{\mathbf{v}}_{k-1} - \mathbf{s}_k\|_2^2 + \mathbb{E}_{k-1}\|\bar{\mathbf{v}}_{k-1} - \bar{\mathbf{v}}_k\|_2^2\right)}{\eta_k M_k} \leq \phi(\bar{\mathbf{v}}_{k-1}) - \mathbb{E}_{k-1}\left[\phi(\bar{\mathbf{v}}_k)\right] + \frac{\eta_k}{cM_k}. \qquad (28)$$

Combining (28) and (26) yields

$$\mathbb{E}_{k-1}\|\nabla\phi_k(\bar{\mathbf{v}}_k)\|^2 \leq 6L\left(\phi(\bar{\mathbf{v}}_{k-1}) - \mathbb{E}_{k-1}\left[\phi(\bar{\mathbf{v}}_k)\right] + 2\frac{\eta_k}{cM_k}\right). \qquad (29)$$

Taking expectation on both sides over all randomness until $\bar{\mathbf{v}}_{k-1}$ is generated and by the tower property, we have

$$\mathbb{E}\|\nabla\phi_k(\bar{\mathbf{v}}_k)\|^2 \leq 6L\left(\mathbb{E}\left[\phi(\bar{\mathbf{v}}_{k-1}) - \phi(\mathbf{v}_*)\right] - \mathbb{E}\left[\phi(\bar{\mathbf{v}}_k) - \phi(\mathbf{v}_*)\right] + \frac{2\eta_k}{cM_k}\right). \qquad (30)$$

Note that $\phi(\mathbf{v})$ is $L$-smooth and hence is $L$-weakly convex, so we have

$$\phi(\bar{\mathbf{v}}_{k-1}) \geq \phi(\bar{\mathbf{v}}_k) + \langle\nabla\phi(\bar{\mathbf{v}}_k), \bar{\mathbf{v}}_{k-1} - \bar{\mathbf{v}}_k\rangle - \frac{L}{2}\|\bar{\mathbf{v}}_{k-1} - \bar{\mathbf{v}}_k\|^2$$

$$= \phi(\bar{\mathbf{v}}_k) + \langle\nabla\phi(\bar{\mathbf{v}}_k) + 2L(\bar{\mathbf{v}}_k - \bar{\mathbf{v}}_{k-1}), \bar{\mathbf{v}}_{k-1} - \bar{\mathbf{v}}_k\rangle + \frac{3}{2}L\|\bar{\mathbf{v}}_{k-1} - \bar{\mathbf{v}}_k\|^2$$

$$\overset{(a)}{=} \phi(\bar{\mathbf{v}}_k) + \langle\nabla\phi_k(\bar{\mathbf{v}}_k), \bar{\mathbf{v}}_{k-1} - \bar{\mathbf{v}}_k\rangle + \frac{3}{2}L\|\bar{\mathbf{v}}_{k-1} - \bar{\mathbf{v}}_k\|^2 \qquad (31)$$

$$\overset{(b)}{=} \phi(\bar{\mathbf{v}}_k) - \frac{1}{2L}\langle\nabla\phi_k(\bar{\mathbf{v}}_k), \nabla\phi_k(\bar{\mathbf{v}}_k) - \nabla\phi(\bar{\mathbf{v}}_k)\rangle + \frac{3}{8L}\|\nabla\phi_k(\bar{\mathbf{v}}_k) - \nabla\phi(\bar{\mathbf{v}}_k)\|^2$$

$$= \phi(\bar{\mathbf{v}}_k) - \frac{1}{8L}\|\nabla\phi_k(\bar{\mathbf{v}}_k)\|^2 - \frac{1}{4L}\langle\nabla\phi_k(\bar{\mathbf{v}}_k), \nabla\phi(\bar{\mathbf{v}}_k)\rangle + \frac{3}{8L}\|\nabla\phi(\bar{\mathbf{v}}_k)\|^2,$$

where (a) and (b) hold by the definition of $\phi_k$.

Rearranging the terms in (31) yields

$$\phi(\bar{\mathbf{v}}_k) - \phi(\bar{\mathbf{v}}_{k-1}) \leq \frac{1}{8L}\|\nabla\phi_k(\bar{\mathbf{v}}_k)\|^2 + \frac{1}{4L}\langle\nabla\phi_k(\bar{\mathbf{v}}_k), \nabla\phi(\bar{\mathbf{v}}_k)\rangle - \frac{3}{8L}\|\nabla\phi(\bar{\mathbf{v}}_k)\|^2$$

$$\overset{(a)}{\leq} \frac{1}{8L}\|\nabla\phi_k(\bar{\mathbf{v}}_k)\|^2 + \frac{1}{8L}\left(\|\nabla\phi_k(\bar{\mathbf{v}}_k)\|^2 + \|\nabla\phi(\bar{\mathbf{v}}_k)\|^2\right) - \frac{3}{8L}\|\nabla\phi(\bar{\mathbf{v}}_k)\|^2$$

$$= \frac{1}{4L}\|\nabla\phi_k(\bar{\mathbf{v}}_k)\|^2 - \frac{1}{4L}\|\nabla\phi(\bar{\mathbf{v}}_k)\|^2$$

$$\overset{(b)}{\leq} \frac{1}{4L}\|\nabla\phi_k(\bar{\mathbf{v}}_k)\|^2 - \frac{\mu}{2L}\left(\phi(\bar{\mathbf{v}}_k) - \phi(\mathbf{v}_*)\right),$$

$$(32)$$

where (a) holds by using $\langle\mathbf{a}, \mathbf{b}\rangle \leq \frac{1}{2}(\|\mathbf{a}\|^2 + \|\mathbf{b}\|^2)$, and (b) holds by the PL property of $\phi$.

Define $\Delta_k = \phi(\bar{\mathbf{v}}_k) - \phi(\mathbf{v}_*)$. Combining (30) and (32), we can see that

$$\mathbb{E}[\Delta_k - \Delta_{k-1}] \leq \frac{3}{2}\left(\mathbb{E}[\Delta_{k-1} - \Delta_k] + \frac{2\eta_k}{cM_k}\right) - \frac{\mu}{2L}\mathbb{E}[\Delta_k],$$

which implies that

$$\left(\frac{5}{2} + \frac{\mu}{2L}\right)\mathbb{E}[\Delta_k] \leq \frac{5}{2}\mathbb{E}[\Delta_{k-1}] + \frac{3\eta_k}{cM_k}.$$

As a result, we have

$$\mathbb{E}[\Delta_k] \leq \frac{5}{5 + \mu/L}\mathbb{E}[\Delta_{k-1}] + \frac{6(\eta_k/cM_k)}{5 + \mu/L} = \left(1 - \frac{\mu/L}{5 + \mu/L}\right)\left(\mathbb{E}[\Delta_{k-1}] + \frac{6\eta_k}{5cM_k}\right)$$

$$\leq \left(1 - \frac{\mu/L}{5 + \mu/L}\right)^k \mathbb{E}[\Delta_0] + \frac{6}{5c}\sum_{j=1}^{k}\frac{\eta_j}{M_j}\left(1 - \frac{\mu/L}{5 + \mu/L}\right)^{k+1-j}.$$

By setting $\eta_k = \eta_0 \exp\left(-\frac{(k-1)}{2}\frac{\mu/L}{5+\mu/L}\right)$, $M_k = \frac{4c}{L\eta_0}\exp\left(\frac{(k-1)}{2}\frac{\mu/L}{5+\mu/L}\right)$ at $k$-th stage, we have

$$\mathbb{E}[\Delta_k] \leq \left(1 - \frac{\mu/L}{5 + \mu/L}\right)^k \mathbb{E}[\Delta_0] + \frac{\eta_0^2 L}{10c^2}\sum_{j=1}^{k}\exp\left(-k\frac{\mu/L}{5 + \mu/L}\right)$$

$$\leq \exp\left(-k\frac{\mu/L}{5 + \mu/L}\right)\Delta_0 + \frac{\eta_0^2 L}{10c^2}k\exp\left(-k\frac{\mu/L}{5 + \mu/L}\right).$$

To achieve $\mathbb{E}[\Delta_K] \leq \epsilon$, it suffices to let $K$ satisfy $\exp\left(-K\frac{\mu/L}{5+\mu/L}\right) \leq \min\left(\frac{\epsilon}{2\Delta_0}, \frac{5c^2\epsilon}{K\eta_0^2 L}\right)$, i.e.

$K \geq \left(\frac{5L}{\mu} + 1\right)\max\left(\log\frac{2\Delta_0}{\epsilon}, \log K + \log\frac{\eta_0^2 L}{5c^2\epsilon}\right)$.

Take $c = \frac{1}{\sqrt{d+3}}$. If $\|\hat{\mathbf{g}}_{1:T_k,i}^k\|_2 \leq \delta \cdot T_k^\alpha$ for $\forall k$, where $0 \leq \alpha \leq \frac{1}{2}$, and note that when $\tau \geq 1$,

$$\max\left(\frac{(\delta + \max_i \|\hat{\mathbf{g}}_{1:\tau,i}^k\|_2)\max(1, 8\tilde{L}^2)}{2c}, c\left(\sum_{i=1}^{d+3}\|\hat{\mathbf{g}}_{1:\tau,i}^k\|_2 + (d + 3)\left(\delta + \max_i \|\hat{\mathbf{g}}_{1:\tau,i}^k\|_2\right)\right)\right)$$

$$\leq \left[(4 + 8\tilde{L}^2)\sqrt{d + 3}\right]\delta \cdot \tau^\alpha$$

so we have $T_k \leq \frac{4c}{L\eta_0}\exp\left(\frac{(k-1)}{2}\frac{\mu/L}{5+\mu/L}\right) \cdot \left[(4 + 8\tilde{L}^2)\sqrt{d + 3}\right]\delta T_k^\alpha$, and hence

$$T_k \leq \left(\frac{4\delta c}{L\eta_0}\exp\left(\frac{(k-1)}{2}\frac{\mu/L}{5 + \mu/L}\right) \cdot \left[(4 + 8\tilde{L}^2)\sqrt{d + 3}\right]\right)^{\frac{1}{1-\alpha}}.$$

Noting that $c = \frac{1}{\sqrt{d+3}}$, we can see that the total iteration complexity is

$$\sum_{k=1}^{K} T_k \leq \left(\frac{4\delta(4 + 8\tilde{L}^2)}{L\eta_0}\right)^{\frac{1}{1-\alpha}} \cdot \frac{\exp\left(K\frac{\mu/L}{(5+\mu/L)(2-2\alpha)}\right) - 1}{\exp\left(\frac{\mu/L}{(5+\mu/L)(2-2\alpha)}\right) - 1} = \widetilde{O}\left(\left(\frac{L\delta^2 d}{\mu^2\epsilon}\right)^{\frac{1}{2(1-\alpha)}}\right).$$

The required number of samples is

$$\sum_{k=1}^{K} m_k = \frac{2(\sigma^2 + C)}{p(1-p)\eta_0^2(d+3)} \cdot \frac{\exp\left(K\frac{\mu/L}{5+\mu/L}\right) - 1}{\exp\left(\frac{\mu/L}{5+\mu/L}\right) - 1} = \widetilde{O}\left(\frac{L^3\sigma^2}{\mu^2\epsilon}\right).$$

$\square$

## A.6 PROOF OF LEMMA 1

*Proof.* For any fixed $\mathbf{w}$, define $(a_{\mathbf{w}}^*, b_{\mathbf{w}}^*) = \arg\min_{a,b}\phi(\mathbf{w}, a, b)$ ($\phi(\mathbf{w}, a, b)$ is strongly convex in terms of $(a, b)$, so the argmin is well-defined and unique). Note that

$$\phi(\mathbf{v}) - \phi(\mathbf{v}_*) = \phi(\mathbf{w}, a, b) - \min_{\mathbf{w},a,b}\phi(\mathbf{w}, a, b) = \phi(\mathbf{w}, a, b) - \phi(\mathbf{w}, a_{\mathbf{w}}^*, b_{\mathbf{w}}^*) + \phi(\mathbf{w}, a_{\mathbf{w}}^*, b_{\mathbf{w}}^*) - \min_{\mathbf{w},a,b}\phi(\mathbf{w}, a, b)$$

We bound $\phi(\mathbf{w}, a, b) - \phi(\mathbf{w}, a_{\mathbf{w}}^*, b_{\mathbf{w}}^*)$ and $\phi(\mathbf{w}, a_{\mathbf{w}}^*, b_{\mathbf{w}}^*) - \min_{\mathbf{w},a,b}\phi(\mathbf{w}, a, b)$ respectively:

- Note that $\phi(\mathbf{w}, a, b)$ is strong convex in $(a, b)$ with modulus $2\min(p, 1 - p)$, so the PL condition holds, which means that

$$\phi(\mathbf{w}, a, b) - \phi(\mathbf{w}, a_{\mathbf{w}}^*, b_{\mathbf{w}}^*) \leq \frac{1}{4\min(p, 1 - p)}\|\nabla_{(a,b)}\phi(\mathbf{w}, a, b)\|^2$$

- 

$$\phi(\mathbf{w}, a_{\mathbf{w}^*}, b_{\mathbf{w}^*}) - \min_{\mathbf{w},a,b} \phi(\mathbf{w}, a, b) = \min_{a,b} \phi(\mathbf{w}, a, b) - \min_{\mathbf{w},a,b} \phi(\mathbf{w}, a, b) \leq \frac{1}{2\mu} \left\| \nabla_{\mathbf{w}} \min_{a,b} \phi(\mathbf{w}, a, b) \right\|^2$$

$$= \frac{1}{2\mu} \left\| \nabla_{\mathbf{w}} \phi(\mathbf{w}, a, b) + \nabla_{\mathbf{w}} \phi(\mathbf{w}, a_{\mathbf{w}}^*, b_{\mathbf{w}}^*) - \nabla_{\mathbf{w}} \phi(\mathbf{w}, a, b) \right\|^2$$

$$\leq \frac{1}{2\mu} \left( 2 \left\| \nabla_{\mathbf{w}} \phi(\mathbf{w}, a, b) - \nabla_{\mathbf{w}} \phi(\mathbf{w}, a_{\mathbf{w}}^*, b_{\mathbf{w}}^*) \right\|^2 + 2 \left\| \nabla_{\mathbf{w}} \phi(\mathbf{w}, a, b) \right\|^2 \right)$$

$$\leq \frac{1}{2\mu} \left( 8\tilde{L}^2 \left\| (a, b) - (a_{\mathbf{w}}^*, b_{\mathbf{w}}^*) \right\|^2 + 2 \left\| \nabla_{\mathbf{w}} \phi(\mathbf{w}, a, b) \right\|^2 \right)$$

$$\leq \frac{1}{2\mu} \left( \frac{8\tilde{L}^2}{4 \min(p^2, (1-p)^2)} \left\| \nabla_{(a,b)} \phi(\mathbf{w}, a, b) \right\|^2 + 2 \left\| \nabla_{\mathbf{w}} \phi(\mathbf{w}, a, b) \right\|^2 \right),$$

where the last inequality holds since $\phi(\mathbf{w}, a, b)$ is strongly convex in $(a, b)$ with modulus $2 \min(p, 1 - p)$.

Combining these two cases, we know that $\phi(\mathbf{v}) - \phi(\mathbf{v}_*) \leq \frac{1}{2\mu'} \|\nabla \phi(\mathbf{v})\|^2$, where $\mu' = \frac{1}{\max\left( \frac{1}{2 \min(p, 1-p)} + \frac{2\tilde{G}^2}{\mu \min(p^2, (1-p)^2)}, \frac{2}{\mu} \right)}$. □

## A.7 AN EXAMPLE THAT SATISFIES PL CONDITION

**One Hidden Layer Neural Network**  One hidden neural network satisfies $h(\mathbf{w}; \mathbf{x}) = \sigma(\mathbf{w}^\top \mathbf{x})$, where $\sigma$ is the activation function. We have the following theorem:

**Theorem 4.** *Let $\sigma$ be the Leaky ReLU activation function such that $\sigma(z) = c_1 z$ for $z > 0$ and $\sigma(z) = c_2 z$ if $z \leq 0$. If $\mathbb{E}[\mathbf{x}|y = 1] = \mathbb{E}[\mathbf{x}|y = -1] = 0$, $\mathbb{E}\left[\mathbf{x}\mathbf{x}'^\top \middle| y = 1, y = -1\right] = \mathbf{0}_{d \times d}$, then*

$$f(\mathbf{w}) := \mathbb{E}_{\mathbf{z}, \mathbf{z}'} \left[ (1 - \sigma(\mathbf{w}^\top \mathbf{x}) + \sigma(\mathbf{w}^\top \mathbf{x}'))^2 \middle| y = 1, y' = -1 \right]$$

*satisfies PL condition with $\mu = 2 \min(c_1^2, c_2^2) \left[ \lambda_{min} \left( \mathbb{E}\left[\mathbf{x}\mathbf{x}^\top \middle| y = 1\right] \right) + \lambda_{min} \left( \mathbb{E}\left[\mathbf{x}\mathbf{x}^\top \middle| y = -1\right] \right) \right]$, where $\lambda_{min}$ stands for the minimum eigenvalue.*

**Remark**: Consider the case that $\mathbf{x}$ is a zero mean Gaussian distribution with non-degenerate convariance matrix. Then $\mu > 0$ since the minimum eigenvalue appeared in the expression of $\mu$ is positive.

*Proof.* Define $g_1(x) = (1-x)^2$, $g_2(\mathbf{w}) = \sigma(\mathbf{w}^\top \mathbf{x}) - \sigma(\mathbf{w}^\top \mathbf{x}')$, $F(\mathbf{w}) = (1 - \sigma(\mathbf{w}^\top \mathbf{x}) + \sigma(\mathbf{w}^\top \mathbf{x}'))^2$. We know that $f(\mathbf{w}) = \mathbb{E}_{\mathbf{z}, \mathbf{z}'} \left[ F(\mathbf{w}) | y = 1, y' = -1 \right]$, $F(\mathbf{w}) = g_1(g_2(\mathbf{w}))$. For fixed $\mathbf{x}$, $\mathbf{x}'$, we can write $\sigma(\mathbf{w}^\top \mathbf{x})$ and $\sigma(\mathbf{w}^\top \mathbf{x}')$ as $a\mathbf{w}^\top \mathbf{x}$ and $b\mathbf{w}^\top \mathbf{x}'$ respectively, and it is obvious that $a^2 \geq \min(c_1^2, c_2^2)$ and $b^2 \geq \min(c_1^2, c_2^2)$. Note that $g_1$ is 2-strongly convex. Since the conditional expectation perserves the strong convexity, as a result, for $\forall \mathbf{w}$, let $\mathbf{w}_p$ be the closest optimal point of

$\mathbf{w}$ such that $f_* = f(\mathbf{w}_p)$, we have

$$f(\mathbf{w}_p) - f(\mathbf{w}) = \mathbb{E}\left[g_1(g_2(\mathbf{w}_p))|y=1, y'=-1\right] - \mathbb{E}\left[g_1(g_2(\mathbf{w}))|y=1, y'=-1\right]$$

$$\geq \mathbb{E}\left[\langle \nabla g_1(g_2(\mathbf{w})), g_2(\mathbf{w}_p) - g_2(\mathbf{w})\rangle|y=1, y=-1\right] + \mathbb{E}\left[(g_2(\mathbf{w}) - g_2(\mathbf{w}_p))^2|y=1, y'=-1\right]$$

$$= \mathbb{E}\left[\langle 2(g_2(\mathbf{w}) - 1), (g_2(\mathbf{w}_p) - g_2(\mathbf{w}))\rangle|y=1, y=-1\right] + \mathbb{E}\left[(g_2(\mathbf{w}) - g_2(\mathbf{w}_p))^2|y=1, y'=-1\right]$$

$$= \mathbb{E}\left[\langle -2(1 - a\mathbf{w}^\top\mathbf{x} + b\mathbf{w}^\top\mathbf{x}'), (a\mathbf{x}^\top - b\mathbf{x}'^\top)(\mathbf{w}_p - \mathbf{w})\rangle|y=1, y'=-1\right]$$
$$\quad + \mathbb{E}\left[\left((a\mathbf{x}^\top - b\mathbf{x}'^\top)(\mathbf{w}_p - \mathbf{w})\right)^2\Big|y=1, y'=-1\right]$$

$$= \mathbb{E}\left[\langle 2(1 - a\mathbf{w}^\top\mathbf{x} + b\mathbf{w}^\top\mathbf{x}')(b\mathbf{x}' - a\mathbf{x}), \mathbf{w}_p - \mathbf{w}\rangle|y=1, y'=-1\right]$$
$$\quad + \mathbb{E}\left[\left((a\mathbf{x}^\top - b\mathbf{x}'^\top)(\mathbf{w}_p - \mathbf{w})\right)^2\Big|y=1, y'=-1\right]$$

$$= \langle \nabla f(\mathbf{w}), \mathbf{w}_p - \mathbf{w}\rangle + \mathbb{E}\left[(\mathbf{w}_p - \mathbf{w})^\top(a\mathbf{x} - b\mathbf{x}')(a\mathbf{x}^\top - b\mathbf{x}'^\top)(\mathbf{w}_p - \mathbf{w})|y=1, y'=-1\right]$$

$$= \langle \nabla f(\mathbf{w}), \mathbf{w}_p - \mathbf{w}\rangle + (\mathbf{w}_p - \mathbf{w})^\top\mathbb{E}\left[(a^2\mathbf{x}\mathbf{x}^\top + b^2\mathbf{x}'\mathbf{x}'^\top)|y=1, y'=-1\right](\mathbf{w}_p - \mathbf{w})$$

$$\geq \langle \nabla f(\mathbf{w}), \mathbf{w}_p - \mathbf{w}\rangle + (\mathbf{w}_p - \mathbf{w})^\top\lambda_{\min}\left(\mathbb{E}\left[(a^2\mathbf{x}\mathbf{x}^\top + b^2\mathbf{x}'\mathbf{x}'^\top)|y=1, y'=-1\right]\right)(\mathbf{w}_p - \mathbf{w})$$

$$\overset{(*)}{\geq} \langle \nabla f(\mathbf{w}), \mathbf{w}_p - \mathbf{w}\rangle + \frac{2\lambda_{\min}\left(\mathbb{E}\left[a^2\mathbf{x}\mathbf{x}^\top|y=1\right]\right) + 2\lambda_{\min}\left(\mathbb{E}\left[b^2\mathbf{x}\mathbf{x}^\top|y=-1\right]\right)}{2}\|\mathbf{w}_p - \mathbf{w}\|^2$$

$$\geq \langle \nabla f(\mathbf{w}), \mathbf{w}_p - \mathbf{w}\rangle + \frac{2\min(c_1^2, c_2^2)\left[\lambda_{\min}\left(\mathbb{E}\left[\mathbf{x}\mathbf{x}^\top|y=1\right]\right) + \lambda_{\min}\left(\mathbb{E}\left[\mathbf{x}\mathbf{x}^\top|y=-1\right]\right)\right]}{2}\|\mathbf{w}_p - \mathbf{w}\|^2$$

$$\geq \min_{\mathbf{w}'}\left[\langle \nabla f(\mathbf{w}), \mathbf{w}' - \mathbf{w}\rangle + \frac{2\min(c_1^2, c_2^2)\left[\lambda_{\min}\left(\mathbb{E}\left[\mathbf{x}\mathbf{x}^\top|y=1\right]\right) + \lambda_{\min}\left(\mathbb{E}\left[\mathbf{x}\mathbf{x}^\top|y=-1\right]\right)\right]}{2}\|\mathbf{w}' - \mathbf{w}\|^2\right]$$

$$= -\frac{1}{4\min(c_1^2, c_2^2)\left[\lambda_{\min}\left(\mathbb{E}\left[\mathbf{x}\mathbf{x}^\top|y=1\right]\right) + \lambda_{\min}\left(\mathbb{E}\left[\mathbf{x}\mathbf{x}^\top|y=-1\right]\right)\right]}\|\nabla f(\mathbf{w})\|^2,$$

where $(*)$ holds since $\lambda_{\min}(A + B) \geq \lambda_{\min}(A) + \lambda_{\min}(B)$, and the last inequality holds since $a^2 \geq \min(c_1^2, c_2^2)$ and $b^2 \geq \min(c_1^2, c_2^2)$. $\qquad\square$

## A.8 DATASET PREPARATION

We construct the datasets in the following ways: For CIFAR10/STL10, we label the first 5 classes as negative ("-") class and the last 5 classes as positive ("+") class, which leads to a 50/50 class ratio. For CIFAR100, we label the first 50 classes as negative ("-") class and the last 50 classes as positve ("+") class. For the imbalanced cases, we randomly remove 90%, 80%, 60% data from negative samples on all training data, which lead to 91/9, 83/17, 71/29 ratio respectively. For testing data, we keep them unchanged.

## A.9 MORE EXPERIMENTS

Model pretraining is effective in many deep learning tasks, and thus we further evaluate the performance of the proposed methods on pretrained models. We first train the model using SGD up to 2000 iterations with an initial step size of 0.1, and then continue training using PPD-SG. We denote this method as *PPD-SG+pretrain* and the results are shown in Figure 2. The parameters are tuned in the same range as in Section 5. It is observed that pretraining model helps the convergence of model and it can achieve the better performance in terms of AUC in most cases.

## A.10 ADDITIONAL EXPERIMENTS WITH DIFFERENT LABELING ORDER

To investigate the effects of labeling order, we also attempt to randomly partition the classes as positive or negative equally. For CIFAR10 and STL10 dataset, we randomly partition the 10 classes into two labels (i.e., randomly select 5 classes as positive label and other 5 classes as negative label). For CIFAR100 dataset, we randomly partition the 100 classes into two labels (i.e., randomly select 50 classes as positive label and other 50 classes as negative label). After that we randomly remove 95%, 90%, from negative samples on all training data, which lead to 20:1, 10:1 ratios respectively. For testing data, we keep them unchanged. We also add AdaGrad for minimizing cross-entropy loss as a new baseline. The corresponding experimental results are included in Figure 3. We can see that PPD-Adagrad and PPD-SG converge faster than other baselines.

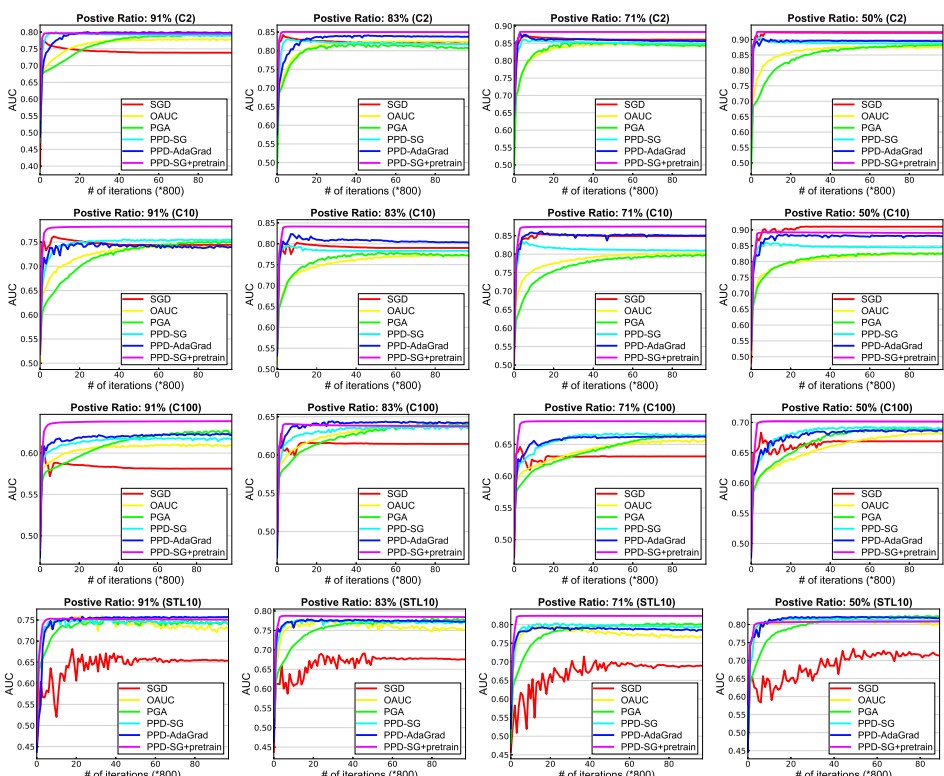

Figure 2: Comparison of testing AUC on Cat&Dog, CIFAR10, CIFAR100 and STL10.

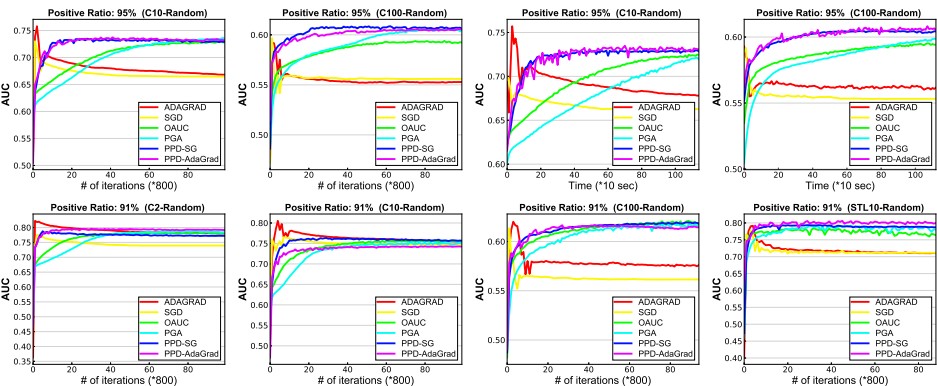

Figure 3: Comparison of testing AUC on Cat&Dog, CIFAR10, CIFAR100 and STL10. For CIFAR10 and STL10 dataset, we randomly partition the 10 classes into two labels (i.e., randomly select 5 classes as positive label and other 5 classes as negative label). For CIFAR100 dataset, we randomly partition the 100 classes into two labels (i.e., randomly select 50 classes as positive label and other 50 classes as negative label).

