# OpenReview forum: "Stochastic AUC Maximization with Deep Neural Networks"
_ICLR.cc/2020/Conference — Accept (Poster)_

### Official Review · AnonReviewer1 · 2019-10-22
**Official Blind Review #1**

**Rating:** 6

**Review:**

This paper proposes two algorithms for the non-convex concave AUC maximization problem, along with theoretical analysis. Experiments show the proposed methods are effective, especially in data imbalanced scenarios.

Strengths:

This paper might be useful and interesting to related research, which overcomes some limitations in previous works such as: 1. the convex assumptions; 2. only considering simple models like linear models; 3. the need of extra memory to store/maintain samples. The proposed method extends existing works to a non-convex setting, which can be applied to deep neural networks, and is applicable for batch-learning and online learning.

The proposed methods achieve better experimental results, especially in the data imbalanced scenarios, which is a real problem that may arise in many scenarios. The paper provides theoretical analysis on the proposed methods, based on Assumption 1, and inspired by the PL condition.

Weaknesses:

I think some comparisons with AdaGrad and related methods should be performed in experiments. Since PPD-AdaGrad is “AdaGrad style”.

The assumptions seem a bit unclear. What does the first assumption in Assumption 1 imply?

Minor Comments:
1. Since the experiments label the first 5/50 classes as negative, and the last 5/50 classes as positive for CIFAR10/CIFAR100, is it possible to provide results on experiments that label in the opposite way (or randomly label 5/50 classes) and add these results in the paper/appendix? Just to make results more convincing and reduce some potential dataset influences.

2. Is it possible to provide some results on more imbalanced positive-negative ratio like 20:1?

3. Is it possible to provide some comparison in terms of actual time, like learning curves with time as x-axis?

4. In the multi-class problems, why are the lower layers shared while last layer separated?

5. Since the extension to multi-classes problems are mentioned in the paper. I like to see some experimental results on this setting.

6. How do the proposed methods perform on models other than NN?

7. I think there is a typo on Page 4, the definition of AUC definition: the latter y should be -1.


**Experience Assessment:**

I do not know much about this area.

**Review Assessment: Checking Correctness Of Derivations And Theory:**

I assessed the sensibility of the derivations and theory.

**Review Assessment: Checking Correctness Of Experiments:**

I assessed the sensibility of the experiments.

**Review Assessment: Thoroughness In Paper Reading:**

I read the paper at least twice and used my best judgement in assessing the paper.

---

> ### Author Response · Authors · 2019-11-15
> **Thank you for your constructive feedback. We have updated the paper accordingly.**
>
> Thank you for your constructive comments.
>
> Q1: I think some comparisons with AdaGrad and related methods should be performed in experiments. Since PPD-Adagrad is “Adagrad style”.
>
> A: To the best of our knowledge, AdaGrad can not be directly applied to solving stochastic AUC maximization problem. So we have included the results when applying AdaGrad for minimizing cross-entropy loss. These results are included in Figure 3 on page 22.
>
>
> Q2: The assumptions seem a bit unclear. What does the first assumption in Assumption 1 imply?
>
> A: The first assumption in Assumption 1 is PL condition on the function $\phi$. It implies that when the gradient of $\phi$ is small, then the objective value is close to the optimal value up to a multiplicative constant $\mu$.
>
>
> Q3: Is it possible to provide results on experiments that label in the opposite way (or randomly label 5/50 classes) and add these results in the paper/appendix? Just to make results more convincing and reduce some potential dataset influences.
>
> A: For CIFAR10 and STL10 dataset, we randomly partition the 10 classes into two labels (i.e., randomly select 5 classes as positive label and other 5 classes as negative label). For CIFAR100 dataset, we randomly partition the 100 classes into two labels (i.e., randomly select 50 classes as positive label and other 50 classes as negative label). We have added a description at Appendix A.10 on page 21 and also included the corresponding results in Figure 3 on page 22.
>
>
> Q4: Is it possible to provide some results on more imbalanced positive-negative ratio like 20:1? Is it possible to provide some comparison in terms of actual time, like learning curves with time as x-axis?
>
> A: We have done more experiments for more imbalanced ratios, i.e., 20:1 and 10:1, and the results are plotted in Figure 3 in the supplement. In particular, in order to create the imbalanced data with 20:1 positive-negative ratio (top four plots), we remove 95% examples with negative label from the original data. The four plots on the bottom are the results with 10:1 positive-negative ratio, for which the data is created by removing 90% examples from the original data. The plot about AUC curve versus actual time is also provided.
>
>
> Q5: In the multi-class problems, why are the lower layers shared while last layer separated?
>
> A: What we meant is that the last layer denotes the classifier and each individual class has a corresponding classifier $h(w_c, x)$. All of these classifiers are built on the same feature induced by the same lower layers.
>
>
> Q6: Since the extension to multi-classes problems are mentioned in the paper. I like to see some experimental results on this setting.
>
> A: Due to time constraint, we are not able to finish this experiment during the rebuttal. We expect to include the results in the final version.
>
>
> Q7: How do the proposed methods perform on models other than NN?
>
> A: We tried linear model but it did not work very well for complex image datasets as used in the experiments.
>
>
> Q8: Typo on Page 4, the definition of AUC definition: the latter y should be -1.
>
> A: Thank you for carefully reading our paper! You are absolutely right and we have fixed this typo in the revision.

---

> > ### Comment · AnonReviewer1 · 2019-11-15
> > **thanks for the clarification**
> >
> > Thanks for the clarification. I am happy to recommend acceptance.
> >
> > There is one more question though, would the PL assumption a bit too restricted since it seems to imply all local optima is not too far away from the global optimum?

---

> > > ### Author Response · Authors · 2019-11-15
> > > **Thank you for your question.**
> > >
> > > Thanks for the insightful question.
> > >
> > > Please note that there is a multiplicative constant $\mu$ in the definition of PL condition. Recall that the definition of PL condition of function $\phi$ is $\phi(v)-\phi(v_*)\leq\frac{1}{2\mu}\|\nabla\phi(v)\|^2$, where $\mu>0$ and $v_*$ is the global minima. If we find a point $v$ that is close to local minima such that $\|\nabla\phi(v)\|\leq \epsilon$, then the objective gap satisfies $\phi(v)-\phi(v_*)\leq\frac{\epsilon^2}{2\mu}$. If $\phi$ is the loss of a deep neural network, $\mu$ is usually very small, so the solution may not be very close to the global minima.
> > >
> > > Finally we would like to mention the following two points.
> > > 1.  PL condition is proved and utilized in some recent neural network theory papers [r1, r2].
> > > 2.  In the setting of deep neural network, the fact that the parameter $\mu$ in PL condition is very small is observed in [r3].
> > >
> > > [r1] Du et al. Gradient descent provably optimizes over-parameterized neural networks. ICLR 2019.
> > > [r2] Allen-Zhu et al. A convergence theory for deep learning via overparameterization. ICML 2019.
> > > [r3] Yuan et al. Stagewise training accelerates convergence of testing error over SGD. NeurIPS 2019.

---

### Official Review · AnonReviewer3 · 2019-10-25
**Official Blind Review #3**

**Rating:** 6

**Review:**

Summary:
The authors propose stochastic algorithms for AUC maximization using a deep neural network. Under the assumption that the underlying function satisfies the PL condition, they prove convergence rates of the proposed algorithms. The key insight is to use the equivalence between AUC maximization and some min-max function. Experiments results show the proposed algorithms works better than some baselines.

Comments:
The technical contribution is to show stochastic optimization algorithms for some kind of min-max functions converge to the optimum under the PL condition. The proposed algorithms have better convergence rates than a naïve application of Rafique et al. The technical results rely on previous work on the PL condition and stochastic optimization of min-max functions. The techniques are not straightforward but not seem to be highly innovative, either.

As a summary, non-trivial algorithms for AUC maximization with neural networks are presented, which could be useful in practice.

Minor Comments:

-How the validation data for tuning parameter are chosen in the experiments? This is absent in the descriptions for experiments.


**Experience Assessment:**

I have published one or two papers in this area.

**Review Assessment: Checking Correctness Of Derivations And Theory:**

I did not assess the derivations or theory.

**Review Assessment: Checking Correctness Of Experiments:**

I did not assess the experiments.

**Review Assessment: Thoroughness In Paper Reading:**

I made a quick assessment of this paper.

---

> ### Author Response · Authors · 2019-11-15
> **Thank you for your valuable comments. We have included our way of choosing validation data.**
>
> Thank you for your insightful comments. We have included the description of the way we chose validation dataset. The revision has been highlighted in red.
>
> Q: How the validation data for tuning parameters are chosen in the experiments? This is absent in the descriptions for experiments.
>
> A: We use 19k/1k, 45k/5k, 45k/5k, 4k/1k training/validation split on C2, C10, C100, and STL10 respectively. We have included this description in Section 5 on page 9.

---

### Official Review · AnonReviewer2 · 2019-10-28
**Official Blind Review #2**

**Rating:** 6

**Review:**

The authors propose two modifications to an algorithm from [Rafique et al 2018] for optimizing AUC under a min-max formulation, prove bounds for the two modifications, and experimentally compare the modifications against SGD and the original algorithm by varying class ratios of four datasets.

The proposal builds on [Rafique et al 2018], so it may be considered incremental. However, the algorithm is carefully analyzed and resulting bounds are stronger. The experimental analysis is fairly minimal, with the proposed modifications performing similarly to the original algorithm from [Rafique et al 2018].

**Experience Assessment:**

I do not know much about this area.

**Review Assessment: Checking Correctness Of Derivations And Theory:**

I did not assess the derivations or theory.

**Review Assessment: Checking Correctness Of Experiments:**

I assessed the sensibility of the experiments.

**Review Assessment: Thoroughness In Paper Reading:**

I read the paper at least twice and used my best judgement in assessing the paper.

---

> ### Author Response · Authors · 2019-11-15
> **Thank you for your review.**
>
> Thank you for your valuable comments and constructive feedback.

---

### Author Response · Authors · 2019-11-15
**General Comments**

Dear reviewers,

Thank you all for your positive ratings and insightful comments. We have updated the paper according to your suggestions. All updates are marked in red. The main summary of the updates are:

1. We have added a description of how we choose training/validation data in our experiments in Section 5 on page 9, as suggested by R2.

2. Per R3’s suggestions, we conducted more experiments on datasets whose classes are randomly partitioned to positive and negative labels with equal size. At the same time, we studied a case with more imbalanced positive-negative ratios (e.g., 20:1, 10:1). The experimental setup and results (both AUC curve versus number of iterations and actual time) are reported in Appendix A.10 and Figure 3.

3. As suggested by R3, we have included AdaGrad as a new baseline in Figure 3.

---

### Decision · Program_Chairs · 2019-12-19

**Decision:**

Accept (Poster)

**Comment:**

The paper proposed using stochastic AUC for dealing with imbalanced data. This paper provides useful insights and experiments on this important problem. I recommend acceptance.